# Study of the Influence of Boundary Conditions on Corneal Deformation Based on the Finite Element Method of a Corneal Biomechanics Model

**DOI:** 10.3390/biomimetics9020073

**Published:** 2024-01-25

**Authors:** Carmelo Gómez, David P. Piñero, Manuel Paredes, Jorge L. Alió, Francisco Cavas

**Affiliations:** 1International School of Doctorate, Technical University of Cartagena, 30202 Cartagena, Spain; gomezgarciacarmelo@gmail.com; 2Department of Optics, Pharmacology and Anatomy, University of Alicante, 03690 Alicante, Spain; david.pinyero@gcloud.ua.es; 3ICA, Université de Toulouse, UPS, INSA, ISAE-SUPAERO, MINES-ALBI, CNRS, 3 rue Caroline Aigle, 31400 Toulouse, France; paredes@insa-toulouse.fr; 4Cornea, Cataract and Refractive Surgery Department, VISSUM, 03016 Alicante, Spain; jlalio@vissum.com; 5Division of Ophthalmology, Department of Pathology and Surgery, Faculty of Medicine, Miguel Hernández University, 03202 Alicante, Spain; 6Department of Structures, Construction and Graphic Expression, Technical University of Cartagena, 30202 Cartagena, Spain

**Keywords:** multizone cornea, constraints influence, patient-specific models, computational time, corneal biomechanics

## Abstract

Implementing in silico corneal biomechanical models for surgery applications can be boosted by developing patient-specific finite element models adapted to clinical requirements and optimized to reduce computational times. This research proposes a novel corneal multizone-based finite element model with octants and circumferential zones of clinical interest for material definition. The proposed model was applied to four patient-specific physiological geometries of keratoconus-affected corneas. Free-stress geometries were calculated by two iterative methods, the displacements and prestress methods, and the influence of two boundary conditions: embedded and pivoting. The results showed that the displacements, stress and strain fields differed for the stress-free geometry but were similar and strongly depended on the boundary conditions for the estimated physiological geometry when considering both iterative methods. The comparison between the embedded and pivoting boundary conditions showed bigger differences in the posterior limbus zone, which remained closer in the central zone. The computational calculation times for the stress-free geometries were evaluated. The results revealed that the computational time was prolonged with disease severity, and the displacements method was faster in all the analyzed cases. Computational times can be reduced with multicore parallel calculation, which offers the possibility of applying patient-specific finite element models in clinical applications.

## 1. Introduction

Biomechanical analyses have allowed for the accuracy of corneal surgeries to be improved as well as for better knowledge of corneal diseases to be obtained by means of developing, extending and applying mechanics to the physiological and physiopathological properties of materials and biological structures [1]. The development of increasingly sophisticated finite element models has been supported by the best knowledge of corneal microstructures [2], obtained with X-ray scattering [3], to allow for the identified risk factors associated with postsurgical problems to be reduced [4] and to avoid serious vision complications.

Keratoconus is a corneal pathology characterized by a multifactorial physiopathology that involves biochemical, genetical and biomechanical components [5]. This disease alters the anatomical integrity of the corneal architecture due to the physical stress exerted by intraocular pressure (IOP). Structural corneal morphology deformation leads to changes in the curvature radii of its surfaces [6], which clinically results in patients’ loss of visual function [7].

The results obtained in the in silico simulation of corneal tissues depend on the implemented material models and parameters [8] based on mechanical tests of similarly behaving human and animal corneas [9]. The material definition is complex due to the multiple layers that compose the cornea and the influence of collagen fibers that modify the isotropic extracellular matrix behavior [10]. The biomechanical cornea response is driven mainly by elastic tissue behavior [11], and analyzing it by hyperelastic models is widely accepted. For this purpose, several material models have been implemented in the developed finite element models [12]. However, the model complexity significantly impacts the computational resolution time of numerical models [13]. Therefore, it is necessary to strike an optimal balance between model fidelity and performance in the specific context where the model is used in clinical practice.

Of all the corneal layers, the stroma, with a nearly 90% thickness, is the main structural layer and is almost 80% water [14]. Approximately two thirds of the collagen fibers are contained within the 45° sectors around the nasal–temporal and superior–inferior directions [15]. In the central corneal zone, the fibers are preferentially oriented in these directions with the same properties [16]. Recent studies conducted by X-ray scattering [2,3] have demonstrated how collagen fibers placed in the frontal circular area, with an approximate diameter of 8 mm centered in the cornea vertex, align mostly in the nasal–temporal and superior–inferior directions. In all, 42% of the fibers show a preferential orientation in the posterior third thickness. This percentage lowers to 22% in the anterior third. The transverse elasticity modulus is independent of depth but heavily depends on the in-plane location [17], whereas the transverse shear stiffness varies with depth, and the shear stiffness is greater in the anterior cornea by one order of magnitude due to the interweaving increment of the lamellas in this zone [18]. The annulus circumscribing the human cornea is located in the posterior third of the limbus thickness, and the fibers are mainly oriented circumferentially in this zone.

The corneal structure has been studied using open-angle X-ray scattering [3]. The results of these research works reveal elastic modulus variation in the meridional and circumferential directions. These variations are attributed to differences in the orientation and concentration of the collagen fibers responsible for the mechanical performance of tissues in different corneal regions. Therefore, the development of biomechanical simulation models with geometric segmentation based on meridional and circumferential sectors is extremely interesting.

Corneal models can be analyzed alone or implemented in global models by considering the influence of other eye components. However, the influence of these components in puff tests is negligible [19]. The generated models are time-consuming and not economical for in silico analyses [20]. 

The boundary conditions must replicate the influence of neighboring tissues on the corneal behavior [21]. When corneal tissue is analyzed alone, the establishment of suitable boundary conditions is a key factor for the results to be accurate, and several options have been extensively used [20]. 

The usual boundary conditions are based on constraints that avoid the displacement and rotation of the nodes placed at the corneo-scleral interface [22] when considering the marked stiffness observed in the limbus zone. However, recent research [8] has reported the importance of introducing a pivoting boundary condition to take into account the limbus influence. Due to limbus stiffness in the circumferential direction, a change in diameter can be neglected. Given this assumption, only limbus rotation contributes to a change in the corneal geometry. In both the embedded and pivoting cases, the displacements of the surrounding tissue contribute to rigid body motion that does not cause stress variation. The corneo-scleral interaction can be contemplated with the use of normal springs in the limbus cross-section. However, this method introduces uncertainties that are difficult to evaluate because the material properties and geometry are not directly quantifiable [8]. Furthermore, the embedded and pivoting boundary conditions are the upper and lower limits of real biomechanical behavior. 

The geometry of the developed finite element model, based on the corneal topography, is affected by the IOP that loads the tissue. The stress–strain distributions in the physiological state are unknown and must be calculated by considering the stress-free geometry (SFG) that is calculated with iterative methods [12,23]. These processes are time-consuming and affect the use of in silico models in real-time clinical applications. However, information about computational times when different iterative methods are applied is lacking, and the variations in the results obtained when using different geometries depend on the degree of disease progression and the boundary conditions that come into play.

The aim of this research was to compare, for the first time, the results obtained with two contour conditions, the so-called embedded and pivoting contour conditions, by considering a finite element model based on a new concept of multiple corneal zones for different degrees of keratoconus disease severity. This comparison was conducted based on using the new FEM model under both contour conditions and at different degrees of disease progression by first comparing two inverse iterative methods to obtain the cornea SFG, i.e., the so-called displacements and prestress methods. Subsequently, the estimated physiological geometry obtained by applying the actual patient-specific IOP to each cornea SFG by the degree of disease progression was also analyzed. For this purpose, four patient-specific corneas affected by keratoconus at different degrees of severity were analyzed. Furthermore, the computational resolution time of the employed numerical methods, namely the displacements and prestress methods, was assessed according to the two established boundary conditions at the limbus, fixed and pivoting, and by considering the degree of keratoconus disease progression. The development of accurate in silico models and shorter computational times are key factors to applying biomechanical models in real-time clinical applications. More precisely in ophthalmology, these methods can be useful for not only characterizing the progression of keratoconus severity but also for making predictions of its clinical course or even for simulating several surgical procedures of this disease.

## 2. Materials and Methods

The material model, finite element description and main hypothesis carried out in this research are outlined below.

### 2.1. Patients

Four finite element models based on real patient-specific cornea models (age range 26–55 years) were developed. All the patients were affected by keratoconus at several degrees of severity according to the Amsler–Krumeich criteria [16]. The specific geometries per patient were measured via a Sirius tomograph (CSO, Florence, Italy). The tomographic examination was randomly performed on one eye of each patient. This involved three complete scans conducted by the same optometrist. The tomographic scan that exhibited the best data acquisition quality index was selected. Table 1 presents the data of the participants in this comparative observational study.

Clinical data were provided by the Vissum Ophthalmological Hospital (Alicante, Spain) and formed part of the IBERIA Biobank (Miguel Hernández University of Elche. Spain). This study was approved by the Ethics Committee of the Technical University of Cartagena, Spain (CEI21_001).

### 2.2. Material Definition

The anisotropic hyperelastic behavior of corneal tissue where viscous loss is negligible and when the cornea is tested in short-term deformation cycles is widely accepted [8]. Corneal tissue is evaluated by considering the continuum mechanics during an isothermal process, where a strain density function is defined.

The position of a material point can be defined by considering an orthonormal Cartesian system, where the coordinates in the initial and deformed configurations can be described by vectors X and x.

Deformation is defined by a deformation tensor $F\;=\frac{\partial x}{\partial X},$ which relates the coordinates in both configurations. The deformation tensor determinant considers the volumetric change in the initial volume. It is expressed as J =det(F). For corneal tissue, it takes a value close to 1 due to its quasi-incompressible behavior [14]. Dividing the deformation into two terms is the usual practice, i.e., one that bears in mind the changes in volume (volumetric deformation) and one that takes into account the variation in shape by maintaining a constant volume (isochoric deformation). The deformation gradient is expressed as $F\;={\;J}^{\frac{1}{3}}\;\bar{F},$ where the volumetric deformation is evaluated with the expression ${\;J}^{\frac{1}{3}}$ and the isochoric deformation is evaluated with the modified material deformation tensor $\bar{F}$.

The strain is evaluated by the right Cauchy–Green deformation tensor, which is evaluated by considering the deformation tensor transpose, denoted as $F^{T}$, and F with expression $C\;={\;F}^{T}F$. The modified right Cauchy–Green deformation tensor is expressed as $\bar{C}={\bar{F}}^{T}\bar{F}$ by contemplating the modified deformation tensor and its transpose. In this research, the strain energy was divided into a volumetric quantity that depends on the Jacobian determinant J and the isochoric component [17].

The isochoric component was divided into an isotropic component by considering the contribution of the matrix and the randomly distributed fibers and an anisotropic component to contemplate the contribution of the preferential fiber orientation according to Equation (1).

The isotropic isochoric component was evaluated with the modified right Cauchy–Green deformation tensor $\bar{C}$, whereas the anisotropic isochoric component was evaluated with the structural tensors ${\;M=m}_{0}⨂m_{0}$ and ${\;N=n}_{0}⨂n_{0},$ where ${\;m}_{0}$ and $n_{0}$ represent the preferential orientations that, according to the X-ray results, were oriented in the nasal–temporal and superior–inferior directions in the central cornea zone and circumferentially in the limbus zone [15]. In this research, the fiber density variation was considered with the zone definition, where different parameters were defined.
(1)$${\psi \;=\;\psi }_{\mathrm{volumetric}}\left(J\right)+{\bar{\psi }}_{\mathrm{Isochoric}}(\bar{C},\;M,\;N)$$

The previously defined equation was evaluated in terms of invariants with the following equation.
(2)$${\psi \;=\;\psi }_{\mathrm{volumetric}}\left(J\right)\;+{\bar{\psi }}_{\mathrm{Isochoric}}({\bar{I}}_{1},{\bar{I}}_{2},{\bar{I}}_{4},{\bar{I}}_{5},{\bar{I}}_{6},{\bar{I}}_{7},{\bar{I}}_{8},{\bar{I}}_{9})$$
where invariants were defined considering the modified right Cauchy–Green deformation tensor and its trace, defined as $\mathrm{tr}\;(\bar{C})$, and the preferential orientations defined by vectors $m_{0}$ and $n_{0}$.
(3)$${\bar{I}}_{1}=\mathrm{tr}\;(\bar{C})$$
(4)$${\bar{I}}_{2}=(\frac{1}{2})\left[{\left(\mathrm{tr}\;(\bar{C})\right)}^{2}\;-\;\mathrm{tr}({\bar{C}}^{2})\right]$$
(5)$${\bar{I}}_{4\;}={\;m}_{0}\bar{C}m_{0}$$
(6)$${\bar{I}}_{5}=m_{0}{\bar{C}}^{2}m_{0}$$
(7)$${\bar{I}}_{6}=n_{0}\bar{C}n_{0}$$
(8)$${\bar{I}}_{7}=n_{0}{\bar{C}}^{2}n_{0}$$
(9)$${\bar{I}}_{8}=m_{0}\bar{C}n_{0}$$
(10)$${\bar{I}}_{9\;}={\left(m_{0}n_{0}\right)}^{2}$$
where ${\bar{I}}_{4}\;$ and ${\bar{I}}_{6}\;$ are the squared fiber stretch, ${\bar{I}}_{5}$ and ${\bar{I}}_{7}$ are related to the transverse fiber deformation, ${\bar{I}}_{8}$ considers the interaction between fiber families and ${\bar{I}}_{9}$ is constant during deformation and is not included in the material deformation description.

For anisotropic behavior, only the influence of parameters ${\bar{I}}_{4}\;$ and ${\bar{I}}_{6}$ was herein considered, as per previously defined research works [17,24].

The general equation to obtain the Cauchy stress tensor σ is based on the partial derivations of the strain energy density function in relation to the invariants.
(11)$$\sigma =\frac{\partial {\;\psi }_{\mathrm{volumetric}}(J)}{\partial J}1+\frac{2}{J}\left[\begin{array}{c} \left(\frac{\partial \bar{\psi }}{\delta {\bar{I}}_{1}}+{\bar{I}}_{1}\frac{\partial \bar{\psi }}{\delta {\bar{I}}_{2}}\right)\bar{b}-\frac{\partial \bar{\psi }}{\delta {\bar{I}}_{2}}{\bar{b}}^{2}+\frac{\partial \bar{\psi }}{\delta {\bar{I}}_{4}}m⨂m+\frac{\partial \bar{\psi }}{\delta {\bar{I}}_{6}}n⨂n\\ -\frac{1}{3}\left(\frac{\partial \bar{\psi }}{\delta {\bar{I}}_{1}}{\bar{I}}_{1}+2\frac{\partial \bar{\psi }}{\delta {\bar{I}}_{2}}{\bar{I}}_{2}+\frac{\partial \bar{\psi }}{\delta {\bar{I}}_{4}}{\bar{I}}_{4}+\frac{\partial \bar{\psi }}{\delta {\bar{I}}_{6}}{\bar{I}}_{6}\right)1 \end{array}\right]$$
where $\bar{b}$ is the modified left Cauchy–Green strain tensor defined as $\bar{b}=\bar{F}{\bar{F}}^{T}$, **m** and **n** are the fiber orientation vectors in the deformed state and **1** is the identity tensor.

The volumetric component $\psi_{\mathrm{volumetric}\;}$ was evaluated according to this equation:(12)$$\psi_{\mathrm{volumetric}\;}=\frac{1}{2}{k_{0}(J\;-\;1)}^{2}$$
where $k_{0}$ is a bulk factor that equals 5.5 MPa herein. The bulk factor was calculated by considering a tangent yield stress value of 0.3 MPa and a Poisson’s ratio of ν=0.49 according to [25] and in the same order as previously implemented [26].

The isochoric isotropic component was evaluated by a 2-parametric Mooney–Rivlin material, where constants $a_{1}$ and $a_{2}$ were obtained from the experimental results [8].
(13)$${\bar{\psi }}_{\mathrm{isotropic}}={\;a}_{1}({\bar{I}}_{1}\;-\;3){+a}_{2}({\bar{I}}_{2}\;-\;3)$$

The isochoric anisotropic component was modeled with the Holzapfel model [24], where constants $k_{1}$ and $k_{2}$ were related to the fiber behavior. The former considers stiffness at less extension, whereas the latter shows the influence at high strain levels. This expression is observed in Equation (13).
(14)$${\bar{\psi }}_{\mathrm{anisotropic}}=\frac{k_{1}}{{2k}_{2}}(\exp[{k_{2}({\bar{I}}_{4}\;-\;1)}^{2}]\;-\;1)+\frac{k_{1}}{{2k}_{2}}(\exp[{k_{2}({\bar{I}}_{6}\;-\;1)}^{2}]\;-\;1)$$

For quasi-incompressible materials, $k_{0}$→∞ and (J - 1) →0, and the product is indeterminate and can be expressed as a Lagrange multiplier, p, and the volumetric term of the Cauchy stress can be expressed as:(15)$$\;\sigma_{\mathrm{volumetric}\;}=k_{0}(J\;-\;1)1\;=-p1$$

The total Cauchy stress tensor can be expressed as:(16)$$\sigma =-p1\;+\frac{2}{J}\left[\begin{array}{c} \left(\frac{\partial \bar{\psi }}{\delta {\bar{I}}_{1}}+{\bar{I}}_{1}\frac{\partial \bar{\psi }}{\delta {\bar{I}}_{2}}\right)\bar{b}-\frac{\partial \bar{\psi }}{\delta {\bar{I}}_{2}}{\bar{b}}^{2}+\frac{\partial \bar{\psi }}{\delta {\bar{I}}_{4}}m⨂m+\frac{\partial \bar{\psi }}{\delta {\bar{I}}_{6}}n⨂n\\ -\frac{1}{3}\left(\frac{\partial \bar{\psi }}{\delta {\bar{I}}_{1}}{\bar{I}}_{1}+2\frac{\partial \bar{\psi }}{\delta {\bar{I}}_{2}}{\bar{I}}_{2}+\frac{\partial \bar{\psi }}{\delta {\bar{I}}_{4}}{\bar{I}}_{4}+\frac{\partial \bar{\psi }}{\delta {\bar{I}}_{6}}{\bar{I}}_{6}\right)1 \end{array}\right]$$

The parameters used in this research for the strain energy density definition were based on the results obtained in previous research works [8,25] and appear in Table 2, where different zones were established to consider the different fiber concentrations observed by X-ray scattering [2,15]. The isotropic isochoric parameters a_1_ and a_2_, which bear in mind the extracellular matrix contribution, were considered to be equal in all the zones, whereas the fiber concentrations were implemented when considering the variation in parameter $k_{1}$, as per previous research works [27,28]. In this research work, the variation in collagen fiber concentrations in terms of the thickness was not considered. Parameter k_2_ was constant according to the hypothesis that the behavior of collagen fibers in large strains is the same in all the corneal zones.

### 2.3. Finite Element Model Definition

The development of patient-specific finite element models starts by modeling the corneal geometry based on topographer measurements [29]. This phase can be summarized in the following steps [30]: Step 1. Point cloud generation: from the Sirius tomograph machine vision algorithm (CSO, Italy), a discrete finite set of spatial points was obtained in polar coordinates (radii and semi meridians), representative of the anterior and posterior corneal surfaces. However, as this process of scanning the algorithm was not complete due to several extrinsic errors, the missing spatial data were obtained by numerical interpolation. Step 2. Geometric reconstruction of corneal surfaces. The corneal surfaces were reconstructed using zonal mathematical functions based on non-uniform rational B-splines (NURBS). Step 3. Solid modeling. By considering the new anterior and posterior corneal surfaces and their corneal thickness, a 3D corneal geometric model was generated. These patient-specific geometrical models have been used to diagnose keratoconus disease [23,31,32].

The first patient-specific cornea model was affected in the incipient phase (G1), and the last three were affected by keratoconus with middle-to-severe grades (G2, G3 and G4).

The corneal models were extended until the limbus zone, where tissue was attached to the sclera. Considering the results of Moore J., et al. [33], an average angle of 40° was taken to define the corneal–scleral interface. Due to existing regional differences in the structural components responsible for the mechanical performance of corneal tissue, a new geometric segmentation concept for the FEM model is herein proposed and is called the multizone corneal model.

As Figure 1 depicts, the corneal geometry is segmented into octant and circumferential sectors by means of cones, with a perpendicular axis to the coronal plane centered on the cornea’s vertex (Figure 1, right side). The sectors at 45° (±22.5°) around the horizontal and vertical meridians were created by considering different fiber concentration zones [15]. The circumferential zones were created to split the cornea into zones of clinical interest [34] by contemplating an additional intermediate value to increase the number of zones. For this purpose, diameters of 2, 4, 6, 8, 10 and 11 mm were employed. The limbus zone was considered from the 11 mm diameter to the external zone [33].

The segmentation process can be summarized in the following steps. In the first step, the cornea was split into octants with transversal planes that contain the vertex (Figure 2a). In the second step, the octants were split into concentrical zones with cones (Figure 2b). The cones were created from sketches and placed in the nasal–temporal direction (Figure 2c), where the diameter was defined by considering the intersection of the cone with a B-spline centered on half of the thickness. The cones’ axes lay perpendicular to the coronal plane through the vertex.

Corneal segmentation allowed for the different strain energy density parameters in each zone to be defined (Figure 3a,b) and the complex corneal models to be developed (Figure 3c). Furthermore, the zones created with corneal segmentation helped to improve the mesh quality (Figure 3d) by allowing regular pattern creation.

Table 2 contains the parameters that defined the different strain energy density zones. These zones can be observed in Figure 3c and were based on previous research results [2,15]. The dark blue zones depict the zones in the nasal–temporal, superior–inferior, central (from the vertex to the 4 mm diameter) and limbus zones, where a high fiber concentration was observed. In the limbus zone, the preferential fiber orientation was circumferential. In the other zones, two fiber families with a preferential orientation were located in the nasal–temporal and superior–inferior directions.

The light green zones in the diagonal octants were defined to consider the zones where the fiber concentration was lower and with a more random distribution. The light blue zones for the transition zones between the different zones were previously defined.

Finite element models (Figure 2d) were implemented with the ANSYS 2020 R1 software (Swanson Analysis System Inc., Houston, PA, USA). For this purpose, 3D elements of a SOLID 186 type with a mixed u-P formulation and a reduced integration were considered. A large displacement method with 10 substeps was used in the solution process.

To accurately obtain the stress distribution, four thickness elements were considered [8], with 11,680 elements in all the corneal models being implemented for comparison purposes. The boundary conditions were applied by means of the definition of multipoint constraints (MPCs) based on the CERIG elements definition. Constraints were applied in the independent node. The IOP was considered similarly to the nodal pressure. The influence of aqueous humor was not taken into account.

### 2.4. Calculation of the Stress and Strain Field in the Measurement Phase

Two inverse iterative methods were used in this study to obtain the SFG: the displacements method [24,35,36] and the prestress method [17,37,38]. The iterative methods implemented in this research are summarized below, where the direction of the previously defined preferential fiber families was assumed to be constant during the iterative process.

#### 2.4.1. Displacements Method

The displacements method (Figure 4) is based on the following steps: The natural physiological geometry was measured with a topographer with the maximum number of iterations and convergence tolerance for the established iterative process (Step 0).

The SFG for the cornea was set (Step 1) and submitted to the IOP (Step 2) to obtain an estimated physiological geometry. The initial geometry measured with the topographer was taken as the initial SFG.

The estimated physiological geometry was compared at the control points with the natural physiological geometry (Step 3). In this research, 57 control nodes were considered. These nodes were placed in the intersections between the sectors and circumferential zones on the faces placed on the anterior and posterior surfaces. Errors were calculated with the maximum norm of the coordinate difference at the control points and compared to the tolerance set in Step 0. In this research, a value of ε = 1·10^−9^ m was considered.

If the difference was lower than the established tolerance, the iterative process stopped and the estimated geometry was considered to be SFG. Otherwise, the iterative process continued until the maximum difference was lower than tolerance or the maximum number of iterations was reached, as previously established. This research work required around 12 iterations for the embedded boundary condition and 14 for the pivoting boundary condition. When SFG was obtained, the physiological IOP was applied and the recovery of the stress–strain fields in the cornea took place (Step 4).

The iterative method was implemented in Python 3.8 (Python Software Foundation), which read and updated the Ansys files (Swanson Analysis System Inc., USA), by considering the flow chart shown in Figure 5.

#### 2.4.2. Prestress Method

The prestress method (Figure 6) was conducted based on the calculation of the stress field that equilibrates the IOP. In a theoretical calculation, when both the stress field and IOP are applied to the finite element model based on the topographic measurement, the obtained displacement level is zero.

The iterative inputs of the iterative process were the natural physiological geometry measured with the topographer, the maximum number of iterations and the displacements tolerance for the established iterative process (Step 0).

The physiological corneal geometry was submitted to the IOP (Step 1), and the gradient deformation tensor was evaluated (Step 2). The total displacements were calculated at 57 control nodes placed at the intersections between the sectors and circumferential zones on the faces placed on the anterior and posterior surfaces.

If the obtained maximum total displacement was lower than the previously established displacements tolerance, the process was stopped and the stress–strain distribution in the physiological state was obtained (Step 3). If displacements level was higher than tolerance, the process continued. For this purpose, the previously obtained gradient deformation tensor was used as the prestress geometry. The process continued until the physiological stress field was obtained with a number of iterations below a previously defined value. This research work required around 14 iterations for the embedded boundary condition and 16 for the pivoting boundary condition. As the SFG could not be obtained directly by this method, an extra step was necessary. It involved applying the stress field calculated to the physiological geometry without the influence of the IOP.

This method was implemented in an iterative process with Python 3.8 (Python Software Foundation), which read and updated the deformation gradient of the previous case until the obtained displacement level was lower than a given tolerance (1·10^−9^ m) when considering the flow chart shown in Figure 7.

### 2.5. Calculation of Displacements and the Stress Field in the Measurement Phase

The strain (m) and stress (Pa) fields in the measurement phase correspond to their estimated physiological configuration (E-P), the values of which were obtained by applying the patient-specific IOP to each SFG for all the pathological conditions under each boundary condition (Figure 8 and Figure 9). These figures show the estimated physiological geometry (iso-color representation) for the inverse iterative displacements and prestress methods by considering both boundary conditions (iso-color embedded vs. pivoting detail) for different degrees of disease progression.

## 3. Results and Discussion

A comparison between both iterative methods was conducted in each pathological scenario and for all the defined boundary conditions. The following subsections show the results obtained for the implemented material model and material properties defined in each zone.

### 3.1. Stress-Free Geometry (SFG)

The stress-free geometries (SFGs) recovered with both iterative methods (displacements and prestress) were obtained by considering all the implemented geometries and boundary conditions. In the prestress method, the initial step involved calculating the physiological stress field. Next, the S-FG was derived by applying the calculated physiological stress field to the corneal finite element model, which was constructed based on the corneal topography.

The results are shown as the total displacements from the measured physiological geometry (see Table 3).

According to the obtained results, the disease progression produced a loss of symmetry, and the obtained maximum displacements were amplified, which were at a maximum for the G4 models. The loss of symmetry was due to keratoconus being a corneal pathology characterized by asymmetry in its corneal structure, which evolves with disease progression. Specifically in the displacement field, a wider geometric variability was observed among the free geometric configurations obtained by each iterative method (Table 3) for the same tolerance calculation level. More precisely, these differences became larger with disease progression due to the mechanical characterization of the material proposed in the numerical model herein presented, which suggests a potential leading role of the elastic modulus in measuring the disease progression risk. These results align with those obtained by other studies [39,40]. Those authors showed that the tangential curvature increased with a decrease in the elastic modulus, which occurred in the corneal structure during keratoconus progression. However, the numerical models of those studies did not use the geometry and patient-specific IOP that depended on the keratoconus severity level. In Figure 10, Figure 11, Figure 12 and Figure 13, the total displacement field (m) obtained for geometries G1 and G3 are shown for both boundary conditions and iterative methods.

The results depicted in Figure 10, Figure 11, Figure 12 and Figure 13 show how the maximum total displacements increased with the degree of disease severity and how these results were affected by the implemented boundary condition and the iterative method. Despite the displacement fields being similar, the maximum displacements varied from one iterative method to another when considering the two boundary conditions. The differences in the maximum displacements were more pronounced with the degree of disease severity.

Differences between iterative methods were observed in the statistical analysis shown in Table 4 when considering the results obtained at the 57 control points placed on the corneal anterior and posterior surfaces. These points were placed at the intersection of the octants with the circumferential zones. According to the obtained results, the difference in the displacements between the iterative methods increased with the degree of disease severity and came very close when the implemented boundary conditions were considered.

For the embedded boundary conditions, the ratio between the maximum total displacements values between the iterative methods showed a mean value of 1.04 (0.02), whereas the mean ratio for the pivoting boundary conditions was 1.07 (0.05). The wider deviations for the pivoting boundary conditions could be attributed to limbus zone irregularity, which hindered the SFG calculation in the models. This factor was more pronounced in the affected corneas with a severe degree of keratoconus and aligns with the results obtained by Moore, J. et al. [33], who analyzed the influence of the limbus on the obtained biomechanical results.

The comparison between the boundary conditions showed differences in the calculated stress-free geometries, which influenced the stress–strain fields recovered when the IOP was applied to recover the physiological state (Table 5). Pandolfi et al. [24] reported that not taking into account the physiological influence of boundary tissues under the boundary conditions of numerical models for biomechanical simulations should be considered an error. Recent research works [8] have demonstrated the importance of incorporating pivoting boundary conditions into cornea analysis by considering the influence of neighboring tissues. However, these works did not take into account the influence of corneal–scleral tissue irregularity when the boundary conditions were imposed, and the variability of the obtained results would depend on disease progression.

The total deformation, observed in inflation tests of whole-globe biomechanics [41], is the sum of the movement as a rigid solid due to the displacements of neighboring tissues and also due to rotation in the corneo-scleral zone. The former does not introduce stress in the corneal tissue, but its influence is a key factor in the results obtained in non-contact tonometry [29], whereas the latter modifies the obtained biomechanical values.

Knowledge of rotational stiffness in corneo-scleral tissue is difficult to obtain and hinders the problem’s solution. However, the real solution is beyond the range of the two analyzed boundary conditions, and the closeness to the results of one or the other depends on the rotational corneo-scleral tissue stiffness, which can be studied in future research works.

### 3.2. Estimated Physiological Geometry (EPG)

The IOP measured in each cornea was applied to the stress-free configurations calculated with both iterative methods and under both boundary conditions to obtain the estimated physiological geometry (EPG). The results are given as the maximum total displacements from the stress-free configuration and are shown in Table 5 and Figure 14, Figure 15, Figure 16 and Figure 17, where the values obtained for geometries G1 and G3 are shown for both boundary conditions and iterative methods.

As observed, the total displacements obtained by the displacements method were equal to those acquired in the SFG configuration because the inverse process was carried out. However, slightly different results were obtained for the prestress method. The reason for this can be attributed to the extra calculation step required to obtain the SFG, where the deformation gradient stress, obtained during the iterative process, was directly applied to the corneal model without applying the IOP. This direct application did not consider updating the stresses according to the material’s anisotropic behavior and modified the acquired results.

From the patient-specific cornea in the incipient state (G1), disease progression produced a loss of symmetry in the obtained displacement field and amplified the bulging effects.

A difference between the iterative methods could be observed in the statistical analysis shown in Table 6 when considering the results obtained at the 57 control points placed on the corneal anterior and posterior surfaces. These points were placed at the intersection of the octants with the circumferential zones. According to the obtained results, the differences in the displacements between the iterative methods came close and were independent of the degree of disease severity.

The obtained displacements were of the same order of magnitude as the values reported by other researchers [25,42,43]. However, these values were based on theoretical models. As far as the authors are aware, this is the first time that displacements have been evaluated for patient-specific corneas with different degrees of keratoconus by two iterative methods and considering the influence of two sets of boundary conditions.

### 3.3. Stress Fields in the Estimated Physiological Geometry

As an application case, the stress distribution in the physiological state was analyzed for both iterative methods by considering the embedded and pivoting boundary conditions. According to the obtained results, both methods showed similar distributions, as depicted in detail in Figure 18, Figure 19, Figure 20 and Figure 21, where a comparison was conducted in the G1 and G3 corneal models. For the patient-specific corneas in the initial disease state, the Von Mises stress values fell within the range of values obtained in other research works [24,25]. However, the values reported by other researchers were based on average properties and theoretical models.

Pandolfi, A. and Manganiello, F. [24] simulated the influence of keratoconus by modifying the material parameters in previously defined zones of a cornea based on average properties. However, patient-specific models were not implemented, and the Von Mises stress evolution in corneas with different degrees of keratoconus was not evaluated. According to the obtained results, the Von Mises stress level showed an increment with the degree of disease progression, and stress concentrations appeared in the zones where the thickness showed reductions or pronounced geometrical variations. The Von Mises stress level reached in the zone nearest the limbus was similar to that obtained previously by other research works for the embedded boundary conditions [33]. The Von Mises stress values associated with numerical peaks were not considered in this comparison.

Introducing pivoting boundary conditions modified the stress field in the limbus zone and demonstrated values depending on the analyzed geometry. For the patient-specific corneas in the initial disease state, introducing the pivoting boundary conditions relaxed the stress level in this zone (Figure 22).

For the patient-specific corneas affected by an advanced degree of keratoconus, the obtained results depended on the geometry and stress concentrations that appeared, which showed the importance of corneal–scleral geometry on the stress values reached in this zone [33]. In the central cornea zone, the results were closer between both the analyzed boundary conditions, and the influence of the boundary conditions was limited (Figure 23).

### 3.4. Strain Fields in the Physiological State

As an application case, the strain distribution in the physiological state was analyzed for both iterative methods by considering the embedded and pivoting boundary conditions. The results are shown in Figure 24, Figure 25, Figure 26 and Figure 27, where a comparison was conducted in the G1 and G3 corneal models. According to the results, both methods showed similar values and strain distributions. In all cases, the maximum values were acquired in the posterior limbus zone, where the boundary conditions were applied. The pivoting condition relaxed the strain concentration in this zone for the corneas in the initial state; however, for the corneas in the incipient disease state and for the advanced keratoconus cases, limbus irregularity led to stress concentrations when the pivoting conditions were introduced. The use of a standardized external geometry with an axisymmetric symmetry and regular thickness could help to eliminate the stress concentration associated with numerical peaks in the limbus zone by focusing the analysis on the central and paracentral zones, which are less affected by the imposed boundary conditions.

By considering the values in the central corneal zones, the degree of disease progression led to an increment in the obtained strain values with a nonlinear relationship with the reached stress field.

### 3.5. Computational Time Analysis

The application of biomechanical models to the medical sector depends not only on the accuracy of the developed methods but also the computational times that must be considered to allow these techniques to be incorporated into real-time applications. As far as the authors know, information about the computational times needed to evaluate corneal biomechanical models and, in particular, to calculate the stress-free configuration is lacking. Elsheikh, A. et al. [35] evaluated the number of iterations needed to calculate this configuration, but they did not analyze the time factor.

A comparison between the methods, which depended on the applied boundary conditions, is shown in Figure 28 and Figure 29 for all the geometries. According to the obtained results, the iterative displacements method was faster than the prestress method for all the analyzed geometries and boundary conditions. The computational time relationship between the prestress and displacements methods was 1.93 (0.17) for the embedded boundary conditions and 1.73 (0.35) for the pivoting boundary conditions.

The extra time required could be attributed to the application of the tensorial stress field that must be applied to all the elements. This process was applied with a factor in each substep and slowed down the advancing iterative process.

In all cases, an extra step was needed to obtain complete mechanical characterization. In the displacements method, where the SFG was obtained, the stress–strain fields in the physiological state resulted when the IOP was applied to the SFG. For the prestress method, where the physiological stress field was calculated, the SFG was obtained in another step by applying the physiological stress field and was calculated according to the corneal finite element model, which was based on corneal topography.

A comparison between the boundary conditions is depicted in Figure 30, where the displacements method was considered. The results showed similar computational times. The time relationship between the pivoting and embedded boundary conditions gave an average value of 1.05 (0.1).

When considering the geometries obtained with the topographer, the computational time increased with degree of keratoconus severity. These geometries indicated abrupt changes in geometry, which made model convergence difficult due to the stress concentration zones.

For clinical applications, a model’s complexity significantly impacts the computational time and is a factor that reduces certainty in predictive performance depending on the clinical application. Therefore, an optimal balance between a realistic model and practical clinical applicability should be sought. In this regard, iterative processes must be accelerated to obtain compatible computational times with normal medical examination times. The influence of the number of core processors researching in parallel was evaluated for all the geometries by considering the pivoting boundary conditions and the displacements method (Figure 31). The results showed that the increment in the number of core processors could accelerate the iterative process by requiring a simulation time that was closer to that of medical examinations. However, these results still cannot be applied during real-time surgeries. As future research lines, new procedures for SFG recovery can be investigated [44,45], and more computational elements can be applied depending on the given study object.

## 4. Conclusions

In this research work, a multizone cornea finite element model was developed for corneal biomechanical analyses due to the growing clinical interest being shown in obtaining a standardized analysis in which different material properties can be applied.

The procedure was implemented in four patient-specific models affected by different degrees of keratoconus severity according to the Amsler–Krumeich criteria, where the stress–strain fields in the physiological state were evaluated based on the SFG calculated by two iterative methods, i.e., the displacements method and the prestress method, with two sets of boundary conditions, i.e., embedded and pivoting conditions.

The obtained results revealed that both iterative methods gave different displacements and stress–strain distributions for the SFG and similar ones for the estimated physiological geometry when considering the same boundary conditions. The obtained maximum values increased with the degree of keratoconus severity. However, the implemented boundary conditions varied the obtained SFG and the recovered stress–strain fields in the physiological state. The differences were more pronounced in the zones nearest the limbus zone, where the pivoting boundary conditions relaxed the stress concentration in the posterior zone of the limbus for healthy corneas and those in the initial keratoconus phase. However, for the corneas in the incipient or advanced state, the stress concentration showed an increment due to limbus irregularity. This factor can be corrected with adaptive geometry by connecting to an axisymmetric zone, which could be the object of future research.

In all the corneas and for all the analyzed boundary conditions, the results were closer in the central and paracentral zones, where the differences between the results were smaller.

The computational time was shorter for the displacements method, and the values obtained for the implemented boundary conditions were very close. The computational time can be reduced by incrementing the number of core processors. The obtained time can be included within the range of medical examination times. All this offers the possibility to apply patient-specific finite element models in clinical applications.

## Figures and Tables

**Figure 1 biomimetics-09-00073-f001:**
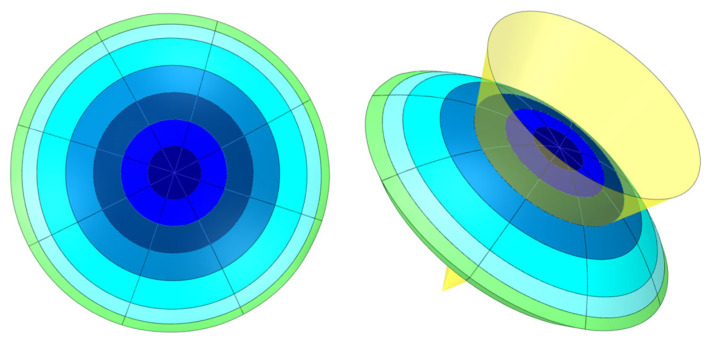
Scheme of the multizone corneal model.

**Figure 2 biomimetics-09-00073-f002:**
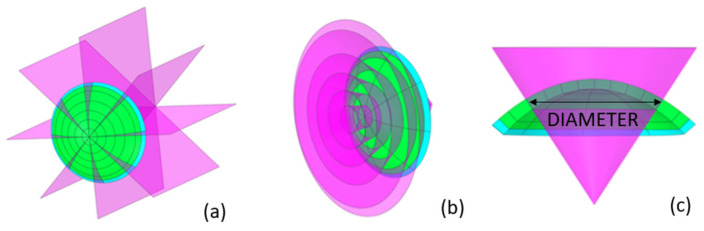
Detail of the corneal segmentation process. (**a**) octant definition; (**b**) circumferential definition; (**c**) detail of diameter definition.

**Figure 3 biomimetics-09-00073-f003:**
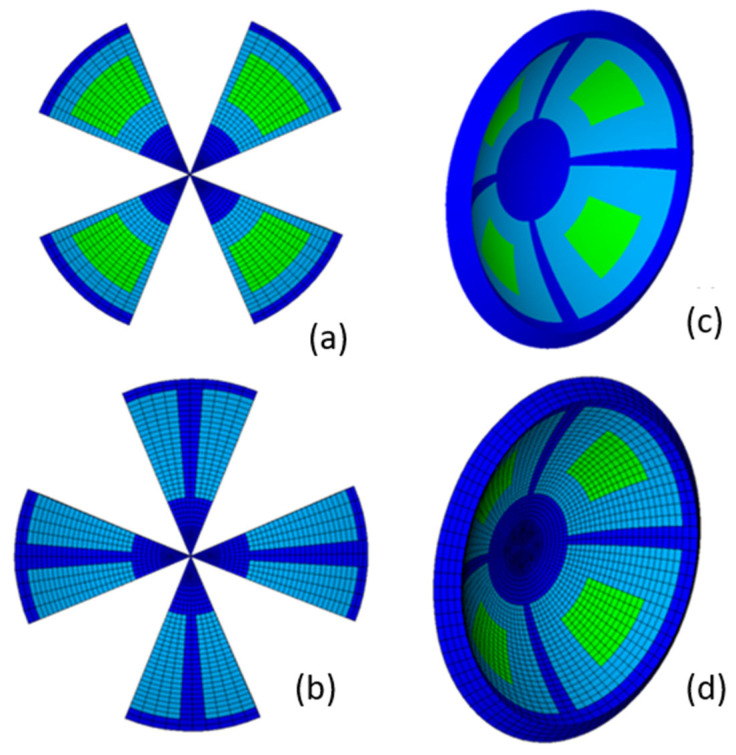
Multizone cornea definition: left side (**a**,**b**): zone definition; right side: detail of the global zone definition (**c**) and detail of the regular mesh pattern (**d**).

**Figure 4 biomimetics-09-00073-f004:**
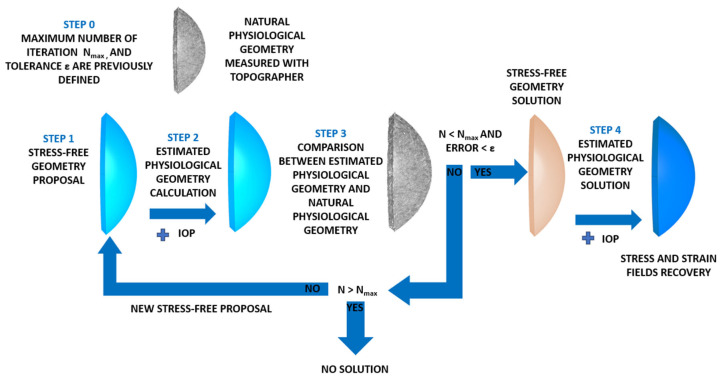
Scheme of the strain–stress field recovery in the physiological state with the displacements method.

**Figure 5 biomimetics-09-00073-f005:**
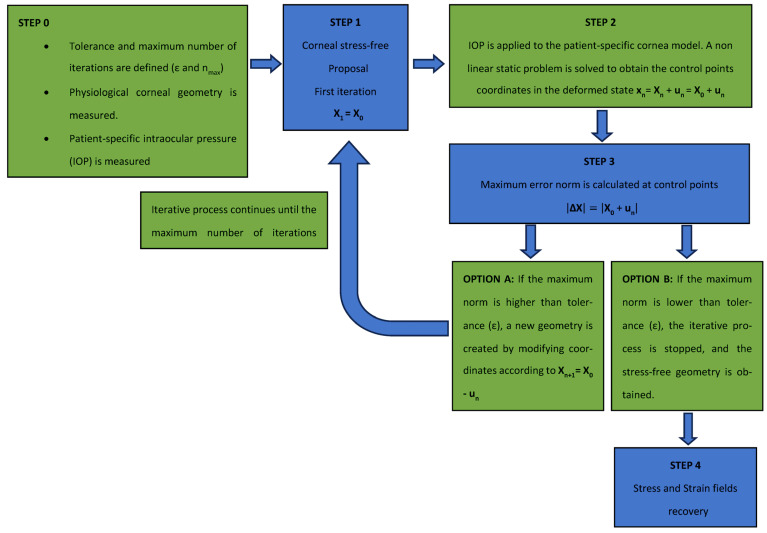
Flow chart of the iterative displacements method process.

**Figure 6 biomimetics-09-00073-f006:**
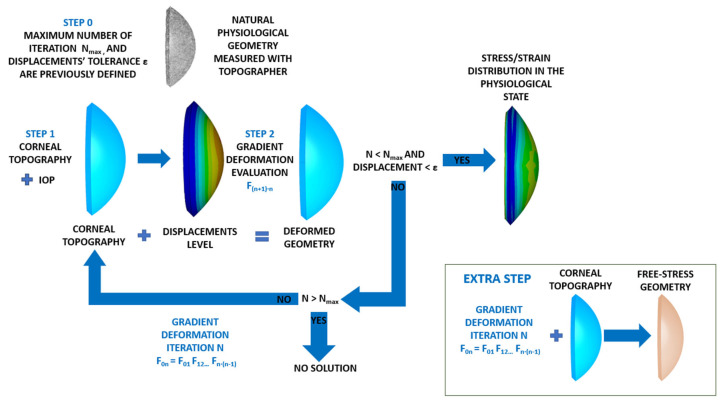
Scheme of the strain–stress field recovery in the physiological state with the prestress method.

**Figure 7 biomimetics-09-00073-f007:**
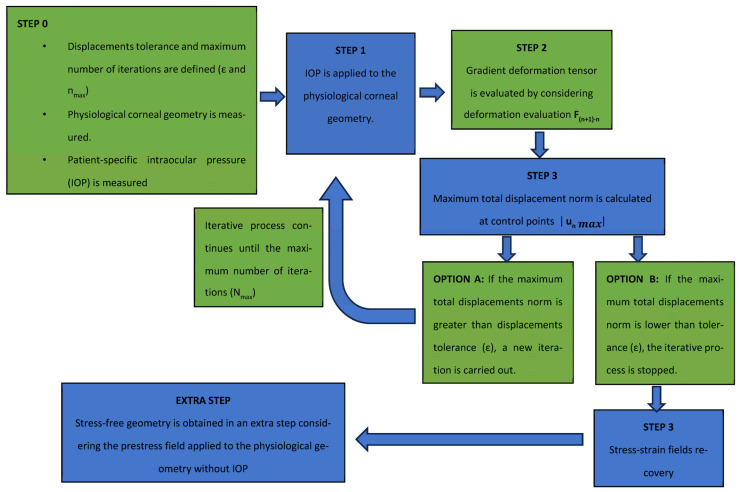
Flow chart of the iterative prestress method process.

**Figure 8 biomimetics-09-00073-f008:**
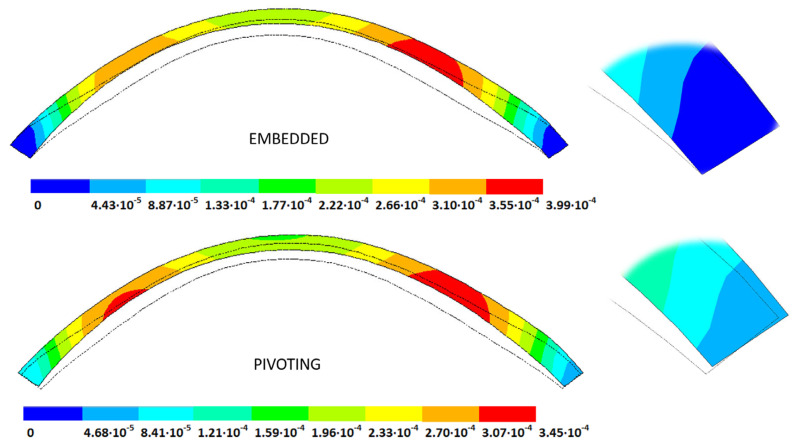
Estimated physiological geometry displacements (m) obtained for a grade III keratoconus cornea under fixed (**top**) and pivoting (**bottom**) boundary conditions. Detail of the boundary condition at the limbus (**top fixed**, **bottom pivoting**). IOP = 14 mmHg applied to the stress-free geometry (solid black line) obtained by the displacements method.

**Figure 9 biomimetics-09-00073-f009:**
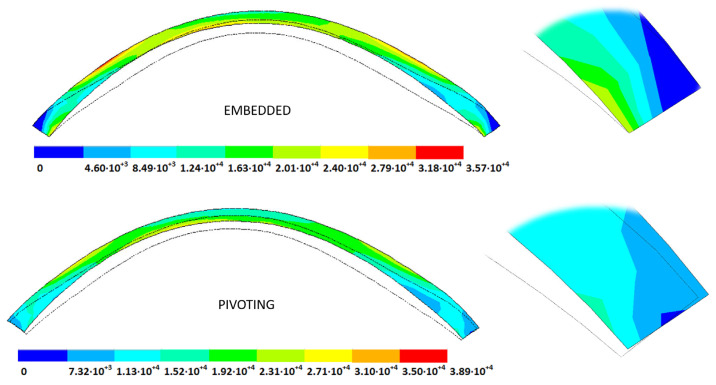
Estimated physiological geometry stresses (Pa) obtained for a grade III keratoconus cornea under fixed (**top**) and pivoting (**bottom**) boundary conditions. Detail of the boundary condition at the limbus (**top fixed**, **bottom pivoting**). IOP = 14 mmHg applied to the stress-free geometry (solid black line) obtained by the displacements method.

**Figure 10 biomimetics-09-00073-f010:**
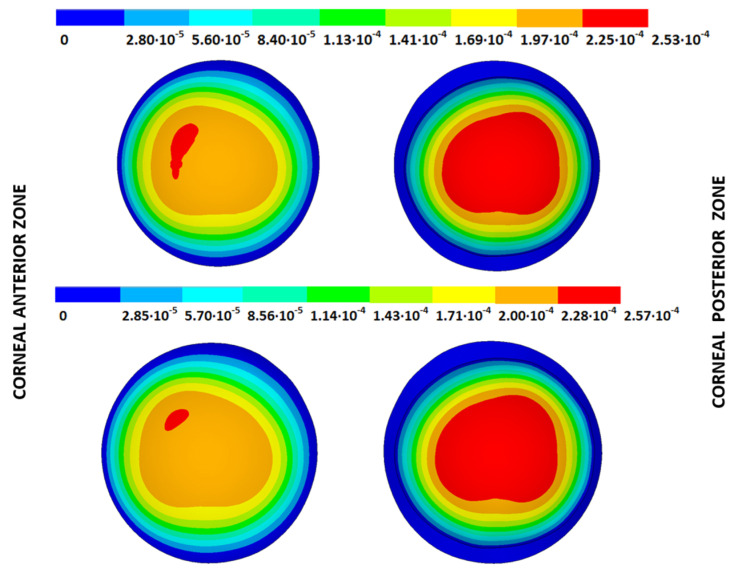
Total distance (m) from the measured physiological geometry to obtain the stress-free geometry. Corneal G1 model (IOP = 15 mmHg). Embedded boundary conditions. (**Upper**: displacements method; **lower**: prestress method).

**Figure 11 biomimetics-09-00073-f011:**
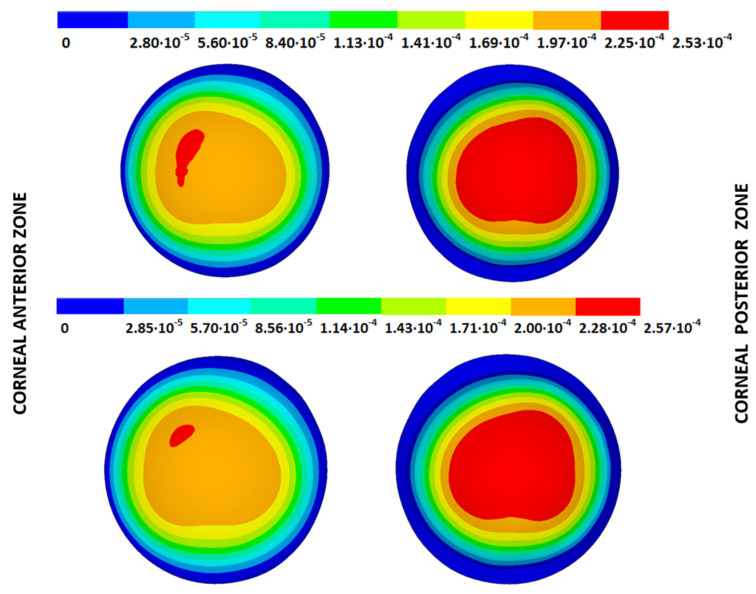
Total distance (m) from the measured physiological geometry to obtain the stress-free geometry. Corneal G1 model (IOP = 15 mmHg). Pivoting boundary conditions. (**Upper**: displacements method; **lower**: prestress method).

**Figure 12 biomimetics-09-00073-f012:**
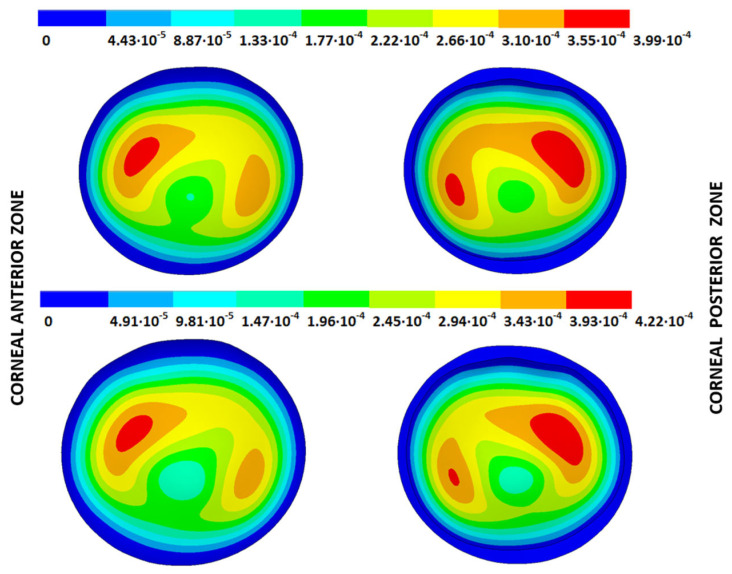
Total distance (m) from the measured physiological geometry to obtain the stress-free geometry. Corneal G3 model (IOP = 14 mmHg). Embedded boundary conditions. (**Upper**: displacements method; **lower**: prestress method).

**Figure 13 biomimetics-09-00073-f013:**
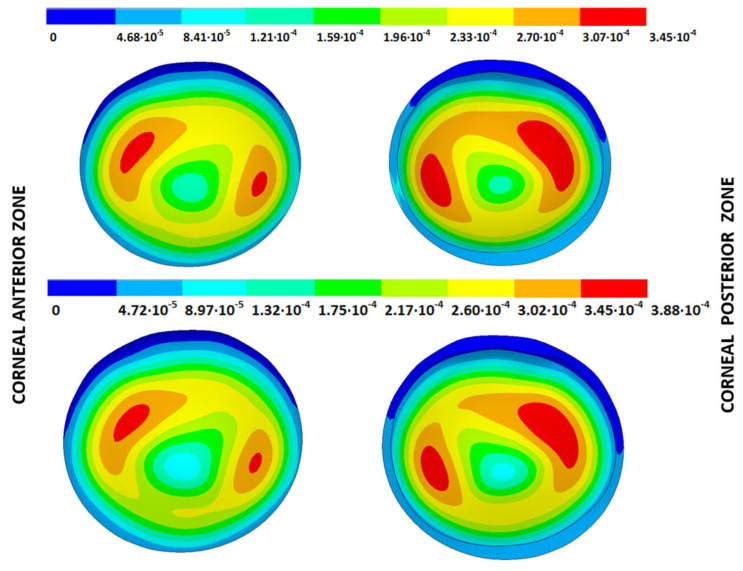
Total distance (m) from the measured physiological geometry to obtain the stress-free geometry. Corneal G3 model (IOP = 14 mmHg). Pivoting boundary conditions. (**Upper**: displacements method; **lower**: prestress method).

**Figure 14 biomimetics-09-00073-f014:**
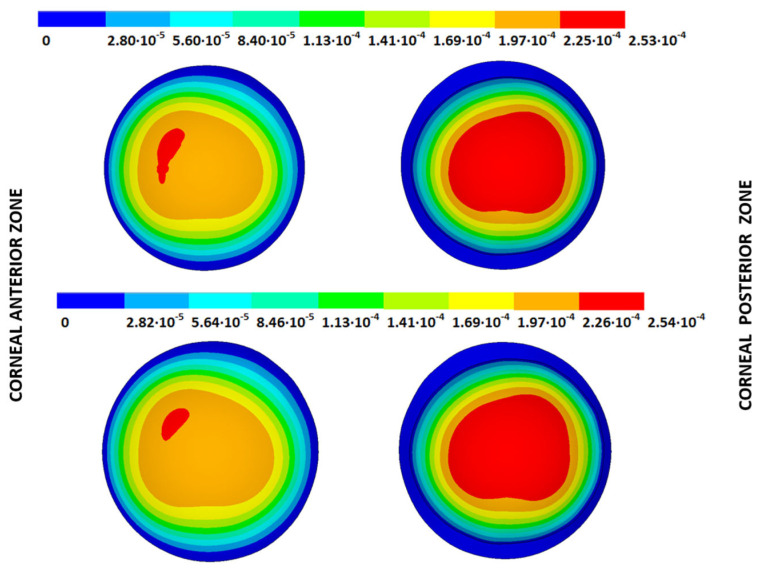
Total distance (m) from the stress-free geometry to obtain the estimated physiological geometry. Corneal G1 model (IOP = 15 mmHg). Embedded boundary conditions. (**Upper**: displacements method; **lower**: prestress method).

**Figure 15 biomimetics-09-00073-f015:**
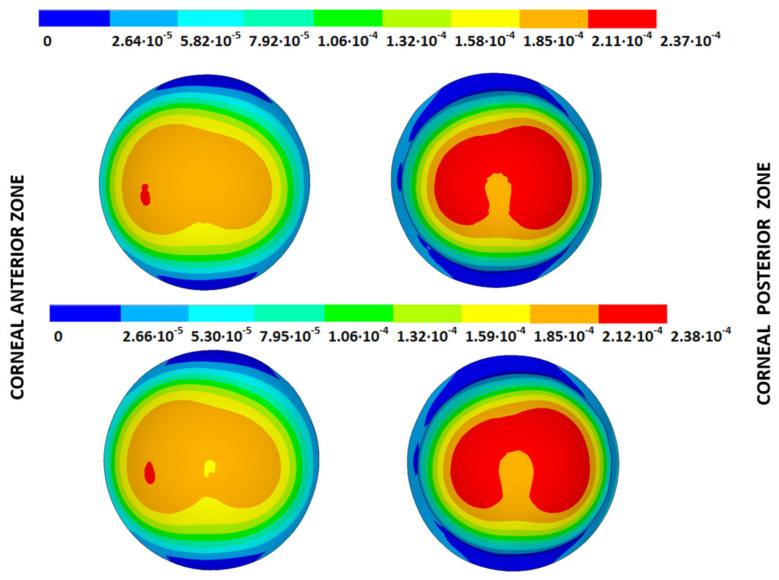
Total distance (m) from the stress-free geometry to obtain the estimated physiological geometry. Corneal G1 model (IOP = 15 mmHg). Pivoting boundary conditions. (**Upper**: displacements method; **lower**: prestress method).

**Figure 16 biomimetics-09-00073-f016:**
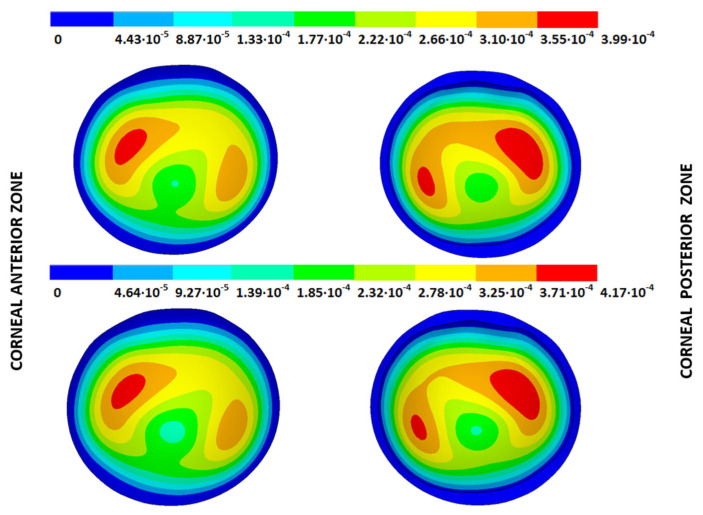
Total distance (m) from the stress-free geometry to obtain the estimated physiological geometry. Corneal G3 model (IOP = 14 mmHg). Embedded boundary conditions. (**Upper**: displacements method; **lower**: prestress method).

**Figure 17 biomimetics-09-00073-f017:**
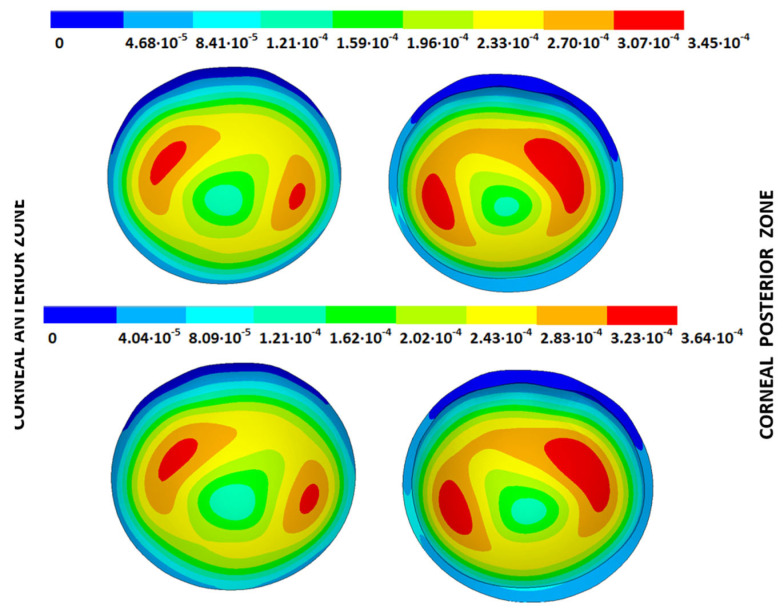
Total distance (m) from the stress-free geometry to obtain the estimated physiological geometry. Corneal G3 model (IOP = 14 mmHg). Pivoting boundary conditions. (**Upper**: displacements method; **lower**: prestress method).

**Figure 18 biomimetics-09-00073-f018:**
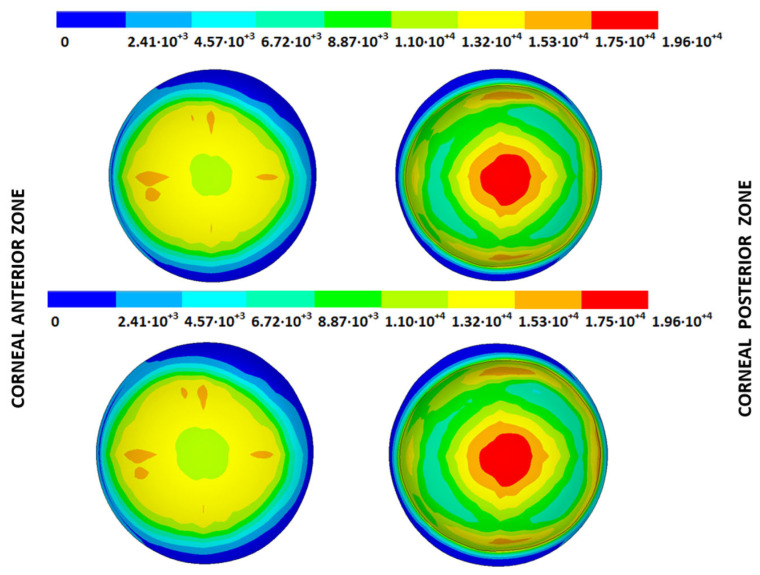
The Von Mises stress field (Pa) for the G1 corneal model (IOP = 15 mmHg) stress-free geometry. Embedded boundary conditions (**Upper**: displacements method; **lower**: prestress method).

**Figure 19 biomimetics-09-00073-f019:**
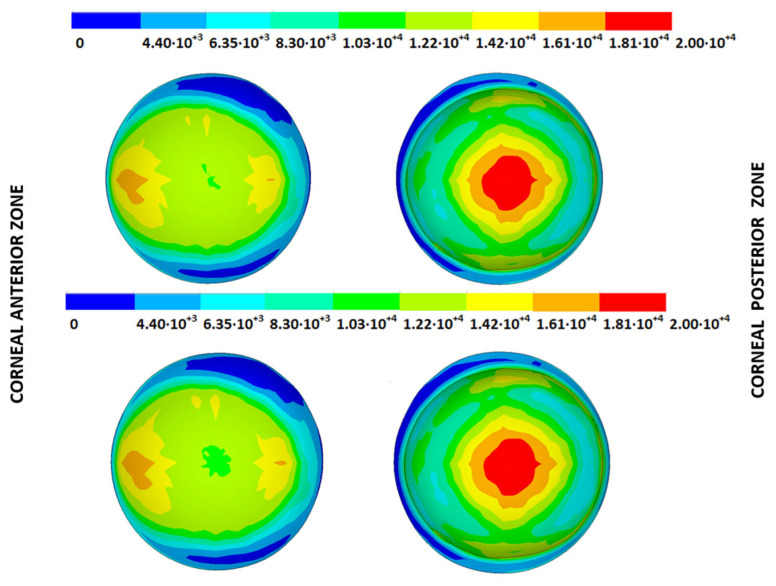
The Von Mises stress field (Pa) for the G1 corneal model (IOP = 15 mmHg) stress-free geometry. Pivoting boundary conditions (**Upper**: displacements method; **lower**: prestress method).

**Figure 20 biomimetics-09-00073-f020:**
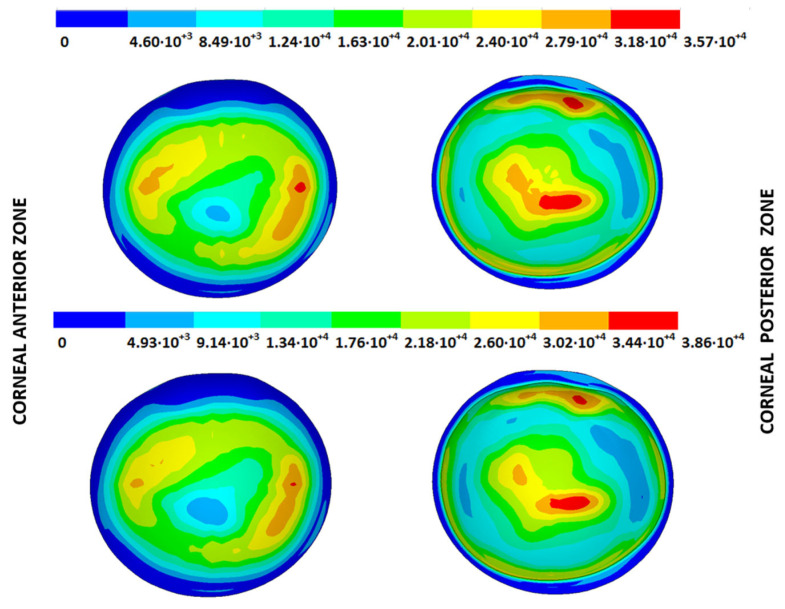
The Von Mises stress field (Pa) for the G3 corneal model (IOP = 14 mmHg) stress-free geometry. Embedded boundary conditions (**Upper**: displacements method; **lower**: prestress method).

**Figure 21 biomimetics-09-00073-f021:**
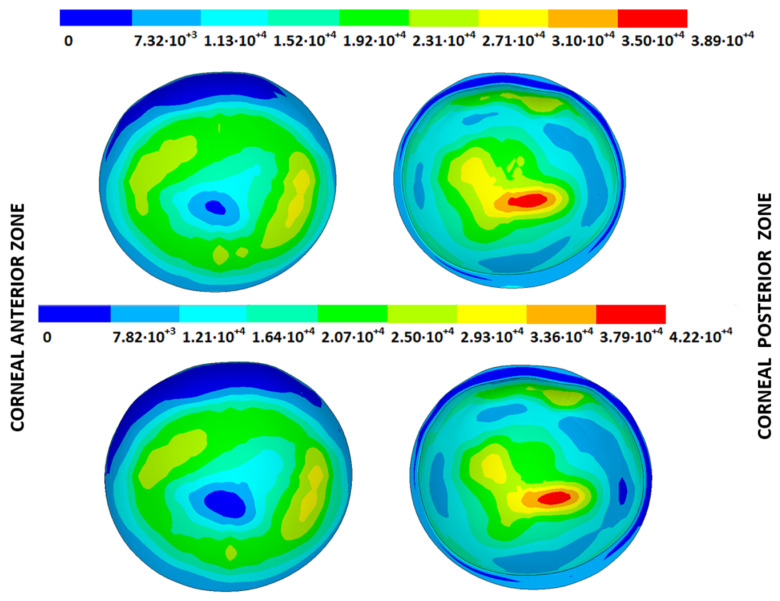
The Von Mises stress field (Pa) for the G3 corneal model (IOP = 14 mmHg) stress-free geometry. Pivoting boundary conditions (**Upper**: displacements method; **lower**: prestress method).

**Figure 22 biomimetics-09-00073-f022:**
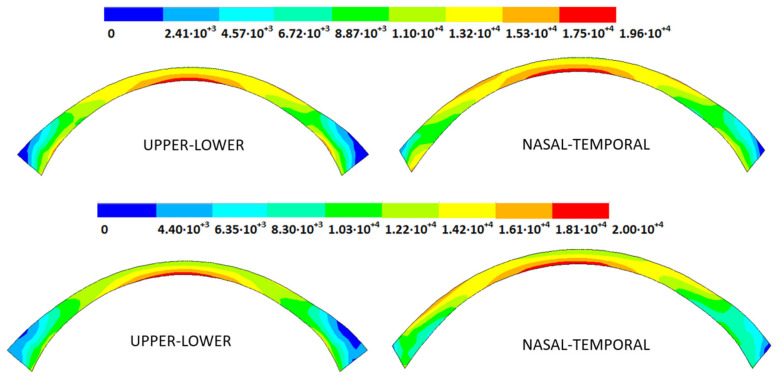
The Von Mises stress field (Pa) for the G1 corneal model (IOP = 15 mmHg) stress-free geometry (**Upper**: embedded boundary condition; **lower**: pivoting boundary condition).

**Figure 23 biomimetics-09-00073-f023:**
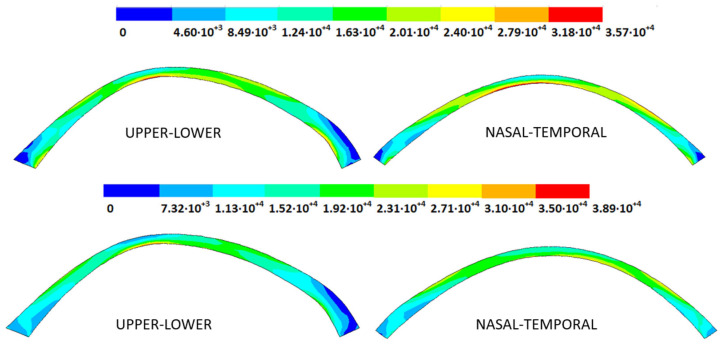
The Von Mises stress field (Pa) for the G3 corneal model (IOP = 14 mmHg) stress-free geometry (**Upper**: embedded boundary condition; **lower**: pivoting boundary condition).

**Figure 24 biomimetics-09-00073-f024:**
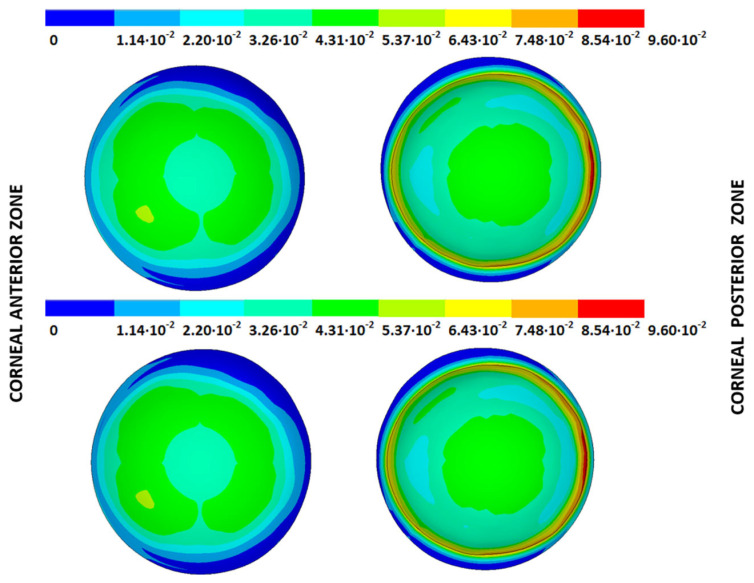
The Von Mises strain field (m/m) for the G1 corneal model (IOP = 15 mmHg) and embedded boundary conditions (**Upper**: displacements method; **lower**: prestress method).

**Figure 25 biomimetics-09-00073-f025:**
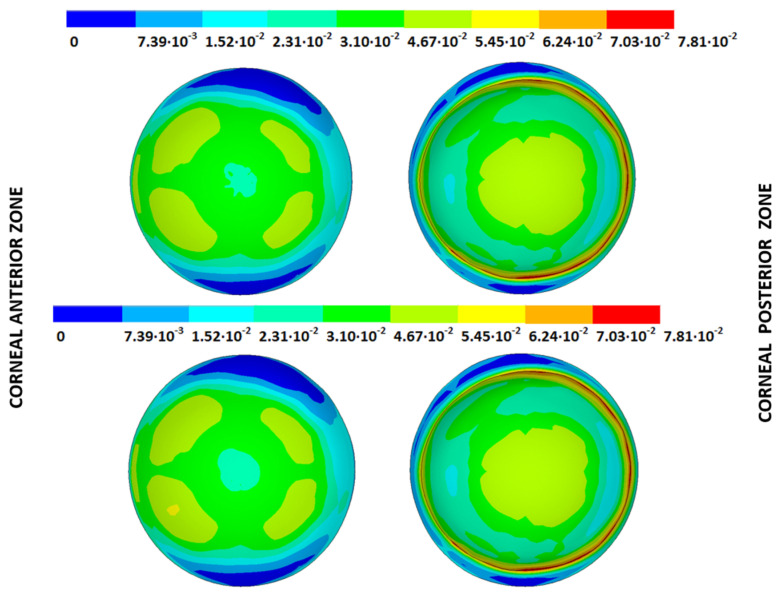
The Von Mises strain field (m/m) for the G1 corneal model (IOP = 15 mmHg) and pivoting boundary conditions. (**Upper**: displacements method; **lower**: prestress method).

**Figure 26 biomimetics-09-00073-f026:**
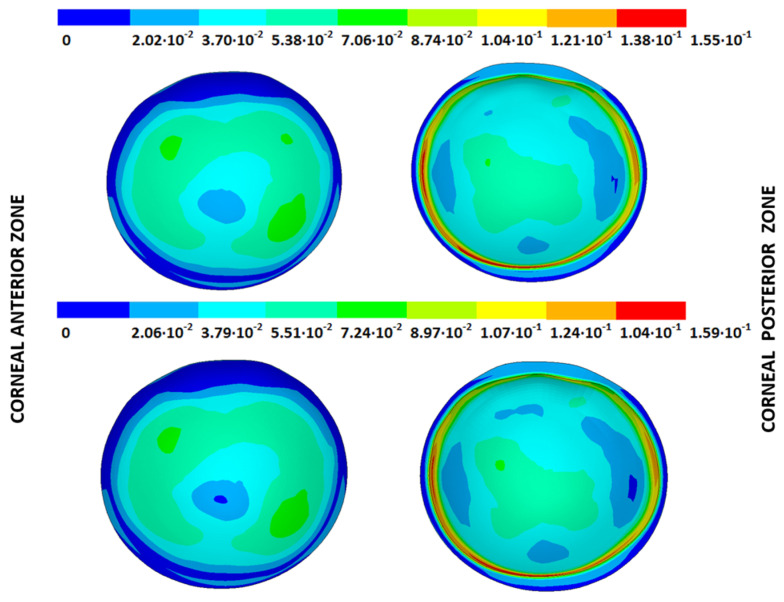
The Von Mises strain field (m/m) for the G3 corneal model (IOP = 14 mmHg) and embedded boundary conditions (**Upper**: displacements method; **lower**: prestress method).

**Figure 27 biomimetics-09-00073-f027:**
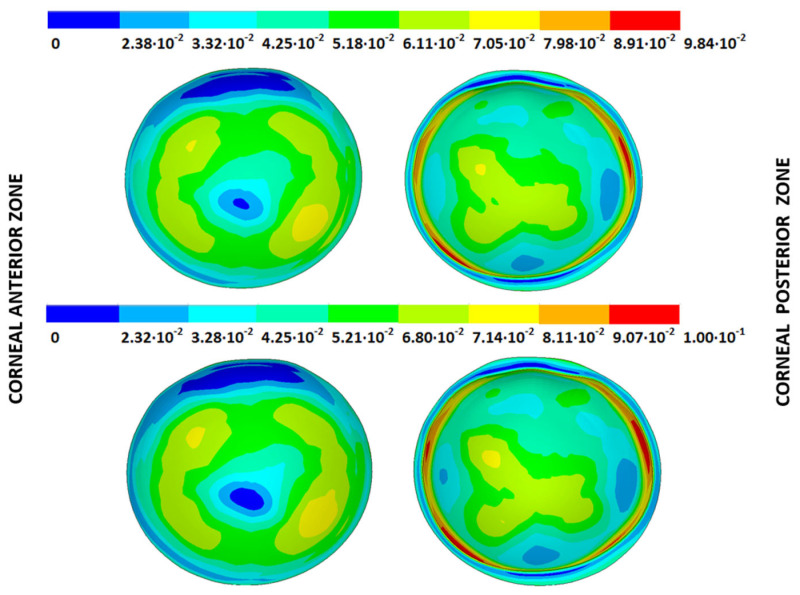
The Von Mises strain field (m/m) for the G3 corneal model (IOP = 14 mmHg) and pivoting boundary conditions (**Upper**: displacements method; **lower**: prestress method).

**Figure 28 biomimetics-09-00073-f028:**
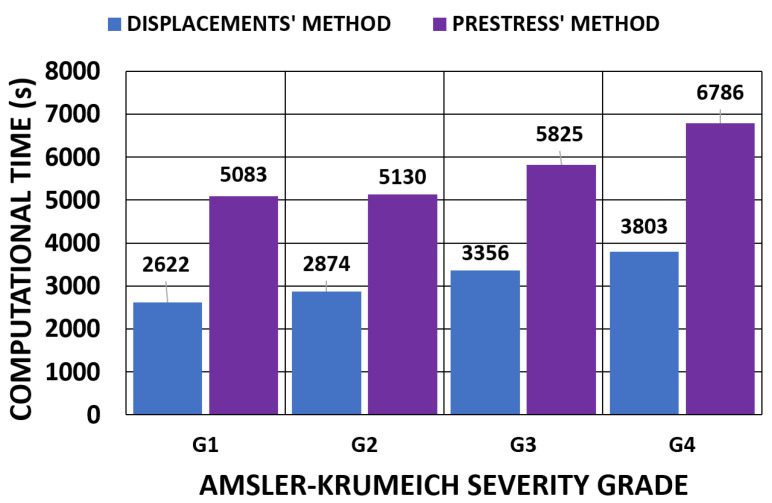
Comparison of the computational time (s) of the iterative process for the embedded boundary condition. (A total of 11,640 SOLID 186 elements with the mixed u-P formulation and reduced integration with large displacements, 10-substep solution and two core processors).

**Figure 29 biomimetics-09-00073-f029:**
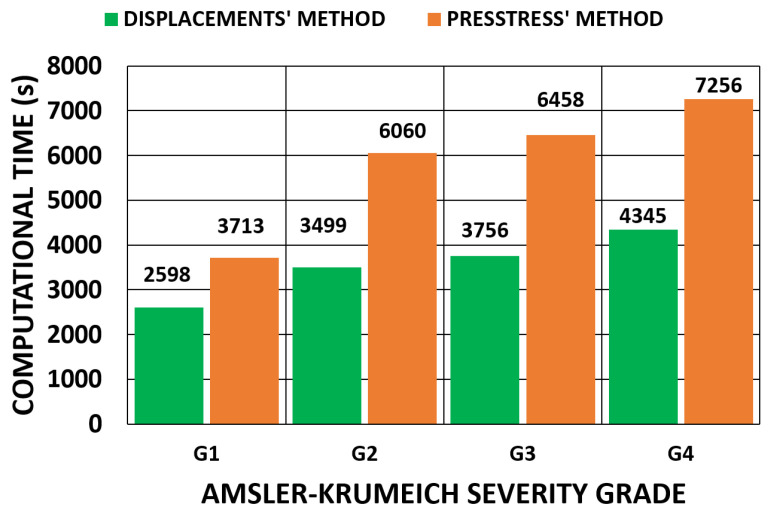
Comparison of the computational time (s) of the iterative process for the pivoting boundary condition. (A total of 11,640 SOLID 186 elements with the mixed u-P formulation and reduced integration with large displacements, 10-substep solution and two core processors).

**Figure 30 biomimetics-09-00073-f030:**
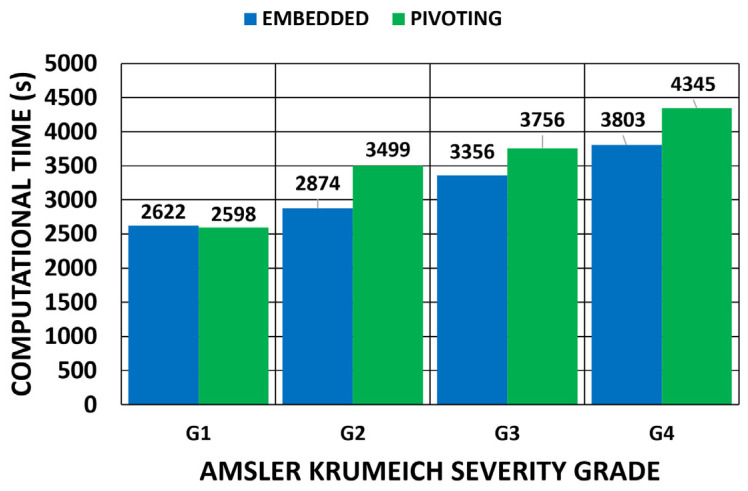
Comparison of computational time (s) depending on the boundary conditions for the displacements method. (A total of 11,640 SOLID 186 elements with the mixed u-P formulation and reduced integration with large displacements and 10-substep solution).

**Figure 31 biomimetics-09-00073-f031:**
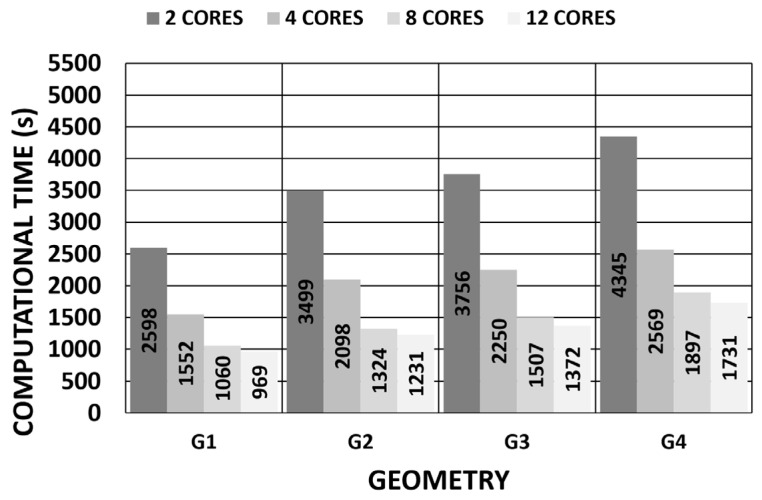
Influence of number of core processors on computational time (pivoting constraint) (11,640 SOLID 186 elements with the mixed u-P formulation and reduced integration with large displacements and 10-substep solution).

**Table 1 biomimetics-09-00073-t001:** Main clinical characteristics of the eyes used during simulations. A-K, Amsler–Krumeich keratoconus grade (G1: grade I; G2: grade II; G3: grade III; G4: grade IV); IOP, intraocular pressure; mean K, mean keratometry; AXL, axial length; CDVA, corrected-distance visual acuity.

A-K	IOP (mm)	Mean K (D)	Age (y)	Gender	Eye	AXL (mm)	Manifest Sphere (D)	Manifest Cylinder (D)	Cylinder Axis (°)	Spherical Equivalent (D)	Decimal CDVA
G1	15	47.29	49	M	OS	21.81	0.5	−0.5	100	0.62	1.05
G2	13	51.59	55	F	OD	24.1	2	−5.5	85	−0.40	0.55
G3	14	53.77	26	F	OS	25.39	−0.5	−2.5	120	−2	0.65
G4	17	69.1	33	M	OD	23.61	0	−2.5	60	−1	0.15

**Table 2 biomimetics-09-00073-t002:** Anisotropic–hyperelastic material parameter definitions.

Material Constants	a_1_ (Pa)	a_2_ (Pa)	k_1_ (Pa)	k_2_ (−)
Central, N-T and S-I zones	40,000	−10,000	50,000	200
Transition zones	40,000	−10,000	37,500	200
Central oblique zones	40,000	−10,000	25,000	200
Limbus	40,000	−10,000	50,000	200

N-T: nasal–temporal, S-I: superior–inferior.

**Table 3 biomimetics-09-00073-t003:** Total distance (m) from the measured physiological geometry to obtain the stress-free geometry.

Stress-Free Geometry (m)		Displacements Method		Prestress Method	
Geometry	IOP (mmHg)	Embedded	Pivoting	Embedded	Pivoting
G1	15	2.53·10^−4^	2.37·10^−4^	2.57·10^−4^	2.41·10^−4^
G2	13	2.87·10^−4^	4.23·10^−4^	2.96·10^−4^	4.35·10^−4^
G3	14	3.99·10^−4^	3.45·10^−4^	4.22·10^−4^	3.88·10^−4^
G4	17	4.78·10^−4^	6.71·10^−4^	4.98·10^−4^	7.42·10^−4^

**Table 4 biomimetics-09-00073-t004:** Statistical analysis of the difference in the total displacements between the iterative displacements method and the iterative prestress method for the calculated stress-free geometry.

G1	Embedded	Pivoting
Maximum (μm)	8.36	7.47
Mean (μm)	2.92	3.07
Standard deviation (μm)	1.83	1.41
**G3**	**Embedded**	**Pivoting**
Maximum (μm)	43.54	43.54
Mean (μm)	13.18	15.59
Standard deviation (μm)	11.35	10.53

**Table 5 biomimetics-09-00073-t005:** Total distance (m) from the stress-free geometry to obtain the estimated physiological geometry.

Estimated Physiological Geometry (m)		Displacements Method		Prestress Method	
Geometry	IOP (mmHg)	Embedded	Pivoting	Embedded	Pivoting
G1	15	2.53·10^−4^	2.37·10^−4^	2.54·10^−4^	2.38·10^−4^
G2	13	2.87·10^−4^	4.23·10^−4^	2.90·10^−4^	4.28·10^−4^
G3	14	3.99·10^−4^	3.45·10^−4^	4.17·10^−4^	3.64·10^−4^
G4	17	4.78·10^−4^	6.71·10^−4^	4.89·10^−4^	7.08·10^−4^

**Table 6 biomimetics-09-00073-t006:** Statistical analysis of the difference in the total displacements between the iterative displacements method and the iterative prestress method for the calculated estimated physiological geometry.

G1	Embedded	Pivoting
Maximum (μm)	0.12	0.14
Mean (μm)	0.04	0.05
Standard deviation (μm)	0.03	0.03
**G3**	**Embedded**	**Pivoting**
Maximum (μm)	0.11	0.11
Mean (μm)	0.04	0.05
Standard deviation (μm)	0.03	0.03

## Data Availability

The data presented in this study are available upon request from the corresponding author.

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
