# Peer review of "Study of the Influence of Boundary Conditions on Corneal Deformation Based on the Finite Element Method of a Corneal Biomechanics Model"

_biomimetics, 2024, doi:10.3390/biomimetics9020073_

Round 1

Reviewer 1 Report

Comments and Suggestions for Authors

The manuscript reports the development of corneal numerical models and their evaluation in relation to different boundary conditions and to the application to four patients physiological geometries of corneas affected by keratoconus. The topic is of interest considered that accurate in-silico models should be potentially useful for in real-time clinical applications including the progression of the severity of the pathology and simulation of surgical procedures. However, to increase the relevance of the manuscript, in particular, the description of the models should be improved for a better understanding of their features also for non-experts of the topic. Also the presentation of the results should be improved, with a more detailed description of the content of the figures and more extended comments.

Detailed comments:

Lines 44-45: The concept expressed in the sentence is not clear, that is how data are related to complication of vision, and should be reviewed.

Line 68 : For clarity, it should better explained how is defined an 8mm area.

Lines 86-88: The use of the expression ‘’structural components’’ should be better explained in relation to mechanical performance.

Lines 106-109: The aim of the study should be described more in detail possibly using more sentences, in order to better highlight the novelty of the content with respect to current literature.

Lines 153-155: The sentence which describes the division of the deformation is not clear and should be reviewed.

Lines 217-218: The content of Table 2 should be better described since the expression ‘’material properties’’ is generic and usually referred to other parameters.

Lines 229-233: The description of the modelling of the corneal geometry is very brief and more details should be provided, possibly using more sentences to better explain each step.

Line 242:  The content of Figure 1 should be described in reference to the two pictures reported.

Lines 242-245: Figure 2 reports pictures relative to different steps of the procedure. The content of each labelled picture should be described, in particular those relative to sectorization (2a, 2b, 2c), which, for clarity should be reported in a separate figure. Also Figure 2d and 2e should be better described better specifying the meaning of ‘’created’’ and ‘’modified’ zones, possibly using more sentences.

Lines 253-255: Light blue zone is not visible in the central zone of Figure 4f and this should be explained.

Line 271: A ‘’red zone’’ is not visible in the figure. Was it referred to Figure 2f?

Lines 292-307: The description of the method is not clear and should be reviewed. The procedure should be described in an orderly way, defining at the beginning of the description the starting condition and then the next steps in reference to the pictures of Figure 3 which should be individually labelled. It is also suggested to use shorter sentences for the description. In addition, for clarity, the content of Table 3 should be reported as a flow chart, for a better understanding of the iterative process.

Lines 316-317 and line 325: The method should be better described in reference to Figure 4, possibly labelling each picture of the figure and referring to individual pictures in the description. In addition, for clarity, the content of Table 4 should be reported as a flow chart.

Line 336: The description of the content of Figure 5 and Figure 6 is missing and should be reported in detail in the text.

Lines 337-346: For clarity, unit of measurements should be reported in the iso-colour representations of Figure 5 and Figure 6. In addition it should be graphically specified to which areas the details are referred to.

Lines 365-366: How can be inferred the loss of the symmetry from data reported in Table 5?

Lines 370-373: The Authors state a potential prognostic role of the elastic modulus in relation to disease progression. However, it is not clear how it is assessed considered that Table 5 reports a total distance. This aspect should be better clarified.

Line 396: In reference to Figures 7-10, a brief comment on the results obtained should be reported in the text.

Lines 404-405: For clarity, it should specified to which parameter the ‘’ratio between maximum values’’ is referred.

Line 416: The content of Table 6 should be described and commented in detail when it is introduced in the text.

Lines 446-457: The results of iso-colour representations for the different condition considered should be commented in the text.

Lines 476-477: Are Von Mises stress values in a healthy state similar to those reported relatively to G1 corneal model? Have these been considered in the study?

Lines 488-489: The sentence is not clear and should be reviewed, possibly splitting the content in more sentences.

Lines 528-531: The sentence is not clear regarding the relationship between the use of a standardized external geometry and stress concentration in the different zones. It is suggested to review it.

Lines 550-551: The sentence seems to be incomplete since main factors which affect the use of biomechanical models in the medical sector are not reported.

Line 560: It is not clear what the parameter ‘’time relation’’ represents and it should be explained.

Lines 606-607: For a better understanding of the relevance of computation time it should be useful an approximate estimate of medical examination times.

Comments on the Quality of English Language

English language should be reviewed in order to improve its fluency. It is suggested to use shorter sentences and to check the use of some words, as an example ‘’comparative’’ and others.

Author Response

Rebuttal/Changes Letter

Dear Reviewer, 

Sincerest thanks for your comments on the manuscript biomimetics-2823120 entitled “Analysis of the influence of boundary conditions on the dis-placement-stress-strain fields of corneas with keratoconus. A comparative study.”. All the changes suggested have been accomplished. In the following lines, a more detailed answer is provided for each question/point. If any additional change is still required, we would appreciate the opportunity to accomplish it. Thank you so much for your help.

Reviewer 1:

1. Reviewer: The manuscript reports the development of corneal numerical models and their evaluation in relation to different boundary conditions and to the application to four patients physiological geometries of corneas affected by keratoconus. The topic is of interest considered that accurate in-silico models should be potentially useful for in real-time clinical applications including the progression of the severity of the pathology and simulation of surgical procedures. However, to increase the relevance of the manuscript, in particular, the description of the models should be improved for a better understanding of their features also for non-experts of the topic. Also the presentation of the results should be improved, with a more detailed description of the content of the figures and more extended comments.

Authors’ response: Thank you very much for your suggestions. The description of the models, the presentation of the results and the description of the figures in the paper have been improved.

2. Reviewer: Lines 44-45: The concept expressed in the sentence is not clear, that is how data are related to complication of vision, and should be reviewed.

Authors’ response: Thank you very much for your suggestion. The concept has been revised.

3. Reviewer: Line 68: For clarity, it should better explained how is defined an 8mm area.

Authors’ response: Thank you very much for your suggestion. The definition of area in the text has been revised.

4. Reviewer: Lines 86-88: The use of the expression ‘’structural components’’ should be better explained in relation to mechanical performance.

Authors’ response: Thank you very much for your suggestion. The concept of structural components has been clarified.

5. Reviewer: Lines 106-109: The aim of the study should be described more in detail possibly using more sentences, in order to better highlight the novelty of the content with respect to current literature.

Authors’ response: Thank you very much for your suggestion. The following text has been introduced:

“The aim of this research is to compare for the first time the results obtained with two contour conditions, the so-called embedded and pivoting contour condition, considering a finite element model based on a new concept of multiple corneal zones, for different degrees of severity of keratoconus disease. This comparison is based on using the new FEM model, under both contour conditions and in different degrees of disease progression, comparing two inverse iterative methods to obtain the stress free-geometry of the cornea, the so-called Displacements´ and Pre-stress´ Methods, and subsequently comparing the estimated physiological-geometry obtained by applying the actual patient-specific IOP to each free-geometry of the cornea by degree of disease progression.”

6. Reviewer: Lines 153-155: The sentence which describes the division of the deformation is not clear and should be reviewed.

Authors’ response: Thank you very much for your suggestion. The sentence has been rewritten.

7. Reviewer: Lines 217-218: The content of Table 2 should be better described since the expression ‘’material properties’’ is generic and usually referred to other parameters.

Authors’ response: Thank you very much for your suggestion. The description of table 2 in the text has been extended.

8. Reviewer: Lines 229-233: The description of the modelling of the corneal geometry is very brief and more details should be provided, possibly using more sentences to better explain each step.

Authors’ response: Thank you very much for your suggestion. The description of geometric modelling has been extended. The following text has been introduced:

“Step 1. Point Cloud Generation: from Sirius tomograph machine vision algorithm (CSO, Italy) a discrete and finite set of spatial points can be obtained in polar coordinates (radii and semi meridians) representative of the anterior and posterior corneal surfaces. However, this process of scanning the algorithm is not complete due to various extrinsic errors, so the missing spatial data is obtained by a numerical interpolation. Step 2. Geometric reconstruction of Corneal Surfaces. Corneal surfaces are reconstructed using zonal mathematical functions based on non-uniform rational B-Splines (NURBS). Step 3. Solid Modeling. Finally taking into account the new anterior and posterior corneal surfaces, and their corneal thickness, a 3D Corneal Geometric-model is generated.”

9. Reviewer: Line 242:  The content of Figure 1 should be described in reference to the two pictures reported.

Authors’ response: Thank you very much for your suggestion. The description of figure 1 in the text has been extended.

10. Reviewer: Lines 242-245: Figure 2 reports pictures relative to different steps of the procedure. The content of each labelled picture should be described, in particular those relative to sectorization (2a, 2b, 2c), which, for clarity should be reported in a separate figure. Also Figure 2d and 2e should be better described better specifying the meaning of ‘’created’’ and ‘’modified’ zones, possibly using more sentences.

Authors’ response: Thank you very much for your suggestions. The description of the different parts of figure 2 have been expanded in the text. Furthermore, the sectorization process has been reported in a separate figure.

11. Reviewer: Lines 253-255: Light blue zone is not visible in the central zone of Figure 4f and this should be explained.

Authors’ response: Thank you very much for your suggestion. Figure 2F has been corrected.

12. Reviewer: Line 271: A ‘’red zone’’ is not visible in the figure. Was it referred to Figure 2f?

Authors’ response: Thank you very much for your suggestions. The color in the text has been corrected, the color in the figure is correct.

13. Reviewer: Lines 292-307: The description of the method is not clear and should be reviewed. The procedure should be described in an orderly way, defining at the beginning of the description the starting condition and then the next steps in reference to the pictures of Figure 3 which should be individually labelled. It is also suggested to use shorter sentences for the description. In addition, for clarity, the content of Table 3 should be reported as a flow chart, for a better understanding of the iterative process.

Authors’ response: Thank you very much for your suggestions. The description of the procedure for obtaining free geometry has been extended and Table 3 modified to create a flow chart. We have numbered the points in to facilitate step by step the calculation procedure.

14. Reviewer: Lines 316-317 and line 325: The method should be better described in reference to Figure 4, possibly labelling each picture of the figure and referring to individual pictures in the description. In addition, for clarity, the content of Table 4 should be reported as a flow chart.

Authors’ response: Thank you very much for your suggestions. The description of the procedure for obtaining free geometry has been extended and Table 3 modified to create a flow chart. We have numbered the points in to facilitate step by step the calculation procedure.

15. Reviewer: Line 336: The description of the content of Figure 5 and Figure 6 is missing and should be reported in detail in the text.

Authors’ response: Thank you very much for your suggestion. The description of figures 5 and 6 has been added to the text.

Figures 5-6 show the estimated physiological geometry (iso-colour representation) for inverse iterative Methods of Displacements and Pre-stress, considering both boundary conditions (iso-colour embedded vs pivoting detail) for different degrees of disease progression.”

16. Reviewer: Lines 337-346: For clarity, unit of measurements should be reported in the iso-colour representations of Figure 5 and Figure 6. In addition it should be graphically specified to which areas the details are referred to.

Authors’ response: Thank you very much for your suggestions. Units have been added in figures’ feet.

17. Reviewer: Lines 365-366: How can be inferred the loss of the symmetry from data reported in Table 5?

Authors’ response: A sentence has been added to clarify the loss of symmetry. Keratoconus is a corneal pathology characterized by an asymmetry in its corneal structure, which progresses with the progression of the disease.

18. Reviewer: Lines 370-373: The Authors state a potential prognostic role of the elastic modulus in relation to disease progression. However, it is not clear how it is assessed considered that Table 5 reports a total distance. This aspect should be better clarified.

Authors’ response: This comment is based on the characterization of the finite element model proposed in this study. Specifically, this model has considered different material parameters for the new multizone concept. The results obtained in the numerical simulation indicate that a greater deformation in the cornea is aligned with the progression of the disease, which suggests a potential diagnostic paper of the elastic module in the measurement of the progression of the disease. A brief explanation has been added to the text.

19. Reviewer: Line 396: In reference to Figures 7-10, a brief comment on the results obtained should be reported in the text.

Authors’ response: Thank you very much for your suggestion. A brief explanation of the results has been added to the text.

20. Reviewer: Lines 404-405: For clarity, it should specified to which parameter the ‘’ratio between maximum values’’ is referred.

Authors’ response: Thank you very much for your suggestion. The concept of ratio between maximum values has been clarified in the text.

21. Reviewer: Line 416: The content of Table 6 should be described and commented in detail when it is introduced in the text.

Authors’ response: Thank you very much for your suggestion. Table 6 content in the text has been expanded.

22. Reviewer: Lines 446-457: The results of iso-colour representations for the different condition considered should be commented in the text.

Authors’ response: Thank you very much for your suggestion. The iso-colour results of the graphic representations have been extended in the text.

23. Reviewer: Lines 476-477: Are Von Mises stress values in a healthy state similar to those reported relatively to G1 corneal model? Have these been considered in the study?

Authors’ response: Sentence has been updated. A healthy state has not been considered in the study.

24. Reviewer: Lines 488-489: The sentence is not clear and should be reviewed, possibly splitting the content in more sentences.

Authors’ response: Thank you very much for your suggestion. The sentence has been rewritten.

25. Reviewer: Lines 528-531: The sentence is not clear regarding the relationship between the use of a standardized external geometry and stress concentration in the different zones. It is suggested to review it.

Authors’ response: Thank you very much for your suggestions. The sentence has been rewritten.

26. Reviewer: Lines 550-551: The sentence seems to be incomplete since main factors which affect the use of biomechanical models in the medical sector are not reported.

Authors’ response: Thank you very much for your suggestion. Thank you very much for your suggestion. The sentence has been completed:

“The application of biomechanical models in the medical sector are dependent not only in the accuracy of the methods developed but also on the integration of these models with other relevant data and technologies [29].”

27. Reviewer: Line 560: It is not clear what the parameter ‘’time relation’’ represents and it should be explained.

Authors’ response: Thank you very much for your suggestion. The concept of time relation has been clarified.

28. Reviewer: Lines 606-607: For a better understanding of the relevance of computation time it should be useful an approximate estimate of medical examination times.

Authors’ response: Thank you very much for your suggestion. However, we cannot give a rough estimate of the time of clinical examination since it depends on each hospital, in this time of examination many factors influence such as medical equipment, auxiliary personnel for clinical examination, etc.

Reviewer 2 Report

Comments and Suggestions for Authors

This paper provides the influence of boundary conditions on the displacement-stress-strain fields of corneas with keratoconus, a corneal disease that alters the shape and structure of the tissue. A novel corneal multizone-based finite element model was established for considering different material properties and fiber orientations in different regions of the cornea. This model was applied to four patient-specific corneas with different severity grades of keratoconus; authors also compared two iterative methods to obtain the stress-free and physiological estimated geometries. In my opinion, this manuscript can be re-considered for publication after dealing with the issues showing below.

1. The title is suggested to change it to “A Study on the Influence of Boundary Conditions on Corneal Deformation Based on Finite Element Method of Corneal Biomechanics Model” or similar expressions.

2. Authors compared the results obtained with two different boundary conditions, embedded and pivoting, to simulate the influence of the neighboring tissues on the corneal behavior1. However, they do not provide a clear justification for choosing these two boundary conditions, nor do they discuss the limitations and assumptions involved in each case. It would be interesting to know how sensitive the results are to the choice of boundary conditions, and whether there are other alternatives that could better represent the realistic corneal-scleral interaction. In addition, they used two inverse iterative methods, the Displacements’ Method and the Pre-stress Method, to obtain the stress-free geometry of the cornea, claiming that both methods produce similar results, but they do not show any quantitative comparison or error analysis to support this claim. It would be useful to have some metrics or criteria to evaluate the accuracy and convergence of the methods, and to quantify the differences between the stress-free geometries obtained with each method.

3. Authors analyze the influence of two types of boundary conditions, embedded and pivoting, on the displacement, stress, and strain fields of the cornea. They observe that the embedded boundary condition produces higher displacements and stresses than the pivoting boundary condition, especially in the posterior limbus zone. They also note that the real solution is somewhere between the two boundary conditions, depending on the rotational stiffness of the corneo-scleral tissue. However, they do not provide any experimental or clinical data to support or validate their assumptions and findings. A more realistic and evidence-based approach to modeling the boundary conditions would be beneficial.

4. The strain values increased with the keratoconus severity grade, but they did not provide any quantitative analysis or statistical comparison of the strain distributions among different grades. 

5. A careful language polishing is needed for the trivial grammar and stylistic errors.

Author Response

Dear Reviewer, 

Sincerest thanks for your comments on the manuscript biomimetics-2823120 entitled “Analysis of the influence of boundary conditions on the dis-placement-stress-strain fields of corneas with keratoconus. A comparative study.”. All the changes suggested have been accomplished. In the following lines, a more detailed answer is provided for each question/point. If any additional change is still required, we would appreciate the opportunity to accomplish it. Thank you so much for your help.

Reviewer 2:

1. Reviewer: This paper provides the influence of boundary conditions on the displacement-stress-strain fields of corneas with keratoconus, a corneal disease that alters the shape and structure of the tissue. A novel corneal multizone-based finite element model was established for considering different material properties and fiber orientations in different regions of the cornea. This model was applied to four patient-specific corneas with different severity grades of keratoconus; authors also compared two iterative methods to obtain the stress-free and physiological estimated geometries. In my opinion, this manuscript can be re-considered for publication after dealing with the issues showing below.

Authors’ response: Thank you very much for your comments.

2. Reviewer: The title is suggested to change it to “A Study on the Influence of Boundary Conditions on Corneal Deformation Based on Finite Element Method of Corneal Biomechanics Model” or similar expressions.

Authors’ response: Thank you very much for your suggestion. Your title proposal has been considered.

3. Reviewer: Authors compared the results obtained with two different boundary conditions, embedded and pivoting, to simulate the influence of the neighboring tissues on the corneal behavior1. However, they do not provide a clear justification for choosing these two boundary conditions, nor do they discuss the limitations and assumptions involved in each case. It would be interesting to know how sensitive the results are to the choice of boundary conditions, and whether there are other alternatives that could better represent the realistic corneal-scleral interaction. In addition, they used two inverse iterative methods, the Displacements’ Method and the Pre-stress Method, to obtain the stress-free geometry of the cornea, claiming that both methods produce similar results, but they do not show any quantitative comparison or error analysis to support this claim. It would be useful to have some metrics or criteria to evaluate the accuracy and convergence of the methods, and to quantify the differences between the stress-free geometries obtained with each method.

Authors’ response: Thank you very much for your suggestions.  The following text has been added in the introduction:

 “Usual boundary conditions are based on constraints that avoid displacement and rotation of nodes placed in the corneo-scleral interface considering the high stiffness observed in the limbus zone. However, recent investigations show the importance of introducing a pivoting boundary condition to take into account the influence of the limbus. Due to the stiffness of the limbus in the circumferential direction, the change of the diameter can be neglected. With this assumption, only the rotation of the limbus contributes to the change of the corneal geometry. In both cases, embedded and pivoting, it is considered that the displacements of the surrounding tissue contribute with a rigid body motion that does not cause a stress variation. The corneo-scleral interaction could be considered with the use of springs normal to the limbus cross-section however this method introduce uncertainties that are difficult to evaluate because materials properties and geometry are not directly quantifiable.”

Statistical analysis have been carried out considering the results obtained in 57 control points showing that both iterative methods produce similar results when the estimated physiological geometry is obtained whereas there is difference in the inverse geometry obtained. This difference is increased with the illness severity grade.

4. Reviewer: Authors analyze the influence of two types of boundary conditions, embedded and pivoting, on the displacement, stress, and strain fields of the cornea. They observe that the embedded boundary condition produces higher displacements and stresses than the pivoting boundary condition, especially in the posterior limbus zone. They also note that the real solution is somewhere between the two boundary conditions, depending on the rotational stiffness of the corneo-scleral tissue. However, they do not provide any experimental or clinical data to support or validate their assumptions and findings. A more realistic and evidence-based approach to modeling the boundary conditions would be beneficial.

Authors’ response: Thank you very much for your suggestions. In this research, embedded and pivoting boundary conditions has been considered the upper and lower limits of the cornea-scleral tissue stiffness. In the introduction the following sentence has been added:

In this way, embedded and pivoting boundary condition can be treated as upper and lower limits of the real behavior.”

The real influence of the corneo-scleral tissue will be study in a further investigation process.

5. Reviewer: The strain values increased with the keratoconus severity grade, but they did not provide any quantitative analysis or statistical comparison of the strain distributions among different grades.

Authors’ response: Thank you very much for your suggestion. The strain values are derived from the displacements. A statistical analysis of displacements has been added to the research showing that differences between iterative methods are negligible in all cases analyzed. Furthermore, the values of the strain and stress fields has been showed as an application case in figures.  Statistical analysis has not been carried out in zones depending on the illness severity grade because the location where these maximum values appear varies depending on the corneal model analysed.

6. Reviewer: A careful language polishing is needed for the trivial grammar and stylistic errors.

Authors’ response: Thank you very much for your suggestion. The text has been reviewed by an expert translator.

Reviewer 3 Report

Comments and Suggestions for Authors

Dear Author(s), 

The manuscript ID: Biomimetics 2024, 9 and titled “Analysis of theinfluence of boundary conditions on the dis-2 placement-stress-strain fields ofcorneas with keratoconus: A comparative study” was reviewed. It is anexperimental paper with already known findings in which actual parametric studyis properly explained, and provides a well-known explanation considering alinear mechanics approach based on the generated testing data.   

The research aim is to apply specific finite element based biomechanicalmodels for surgeries applications. Findings from the literature search werediscussed, and the conclusion was consistent with generated data. 

It was evaluated that it is an experimental paper supported with FEM andfinding could attract readers especially modeling and analysis persons in thestate of the art, practisinary and research engineers.  It may be considered to publish in Biomimetics(Basel) Journal after some revisions as:  

1. Please provide some distint differences between displacement and strainmeasurements considering FEM.   

2. Please explain how to find orwhich parameters are considered during defining the boundary conditions.   

3. Please mention and provide some information about mesh size and geometryselections.   

4. Provide some information on convergence or divergence between measureddata and biomechanical model.    

My best regards, 

Author Response

Dear Reviewer, 

Sincerest thanks for your comments on the manuscript biomimetics-2823120 entitled “Analysis of the influence of boundary conditions on the dis-placement-stress-strain fields of corneas with keratoconus. A comparative study.”. All the changes suggested have been accomplished. In the following lines, a more detailed answer is provided for each question/point. If any additional change is still required, we would appreciate the opportunity to accomplish it. Thank you so much for your help.

Reviewer 3:

1. Reviewer: The manuscript ID: Biomimetics 2024, 9 and titled “Analysis of the influence of boundary conditions on the dis-2 placement-stress-strain fields of corneas with keratoconus: A comparative study” was reviewed. It is an experimental paper with already known findings in which actual parametric study is properly explained, and provides a well-known explanation considering a linear mechanics approach based on the generated testing data. The research aim is to apply specific finite element based biomechanical models for surgeries applications. Findings from the literature search were discussed, and the conclusion was consistent with generated data. It was evaluated that it is an experimental paper supported with FEM and finding could attract readers especially modeling and analysis persons in the state of the art, practisinary and research engineers. It may be considered to publish in Biomimetics(Basel) Journal after some revisions as: .

Authors’ response: Thank you very much for your comments.

2. Reviewer: Please provide some distinct differences between displacement and strain measurements considering FEM.

Authors’ response: Displacements are the variation expressed in meters of the coordinates of the nodes of the corneal finite element model when it is submitted to boundary condition and loads whereas strain measurements are calculated dividing the distance between two points in the deformed state by the distance in the initial state (nondimensional).

3. Reviewer: Please explain how to find or which parameters are considered during defining the boundary conditions.

Authors’ response: Boundary conditions are applied when the finite element model is created to the nodes located in the limbus zone. In these nodes, for the embedded boundary conditions the displacements and rotations are constrained avoiding the movement whereas for the pivoting boundary condition the following procedure has been carried out: Extra nodes has been created in middle of the limbus thickness in each section where there is nodes. The extra node is connected to the nodes of the limbus with multipoint constraints of type CERIG.  Only the displacements are constrained in the extra node (Rotations are free) allowing the rotation around the middle line of the limbus.

4. Reviewer: Please mention and provide some information about mesh size and geometry selections.

Authors’ response: Thank you very much for your suggestions. The number of total elements has been added:

Finite element models (Figure 2g) were implemented with ANSYS software (Swanson Analysis System Inc. USA). For this purpose, 3D elements of type SOLID 186 with mixed u-P formulation and reduced integration were considered. Large displacement method with 10 sub steps has been used in the solution process.

In order to obtain the distribution of the stress with accuracy, four elements in the thickness have been considered [8] with a total of 11680 elements in all corneal models implemented for comparison purposes. Boundary conditions are applied by means of the definition of multipoint constraints (MPCs) based on the definition of CERIG elements. Constraints are applied in the independent node. The intraocular pressure has been considered like a nodal pressure. The influence of aqueous humor has not been considered”.

5. Reviewer: Provide some information on convergence or divergence between measured data and biomechanical model.

Authors’ response: Thank you very much for your suggestion. Statistical tables have been added showing the convergence of measured data and biomechanical model.

Round 2

Reviewer 2 Report

Comments and Suggestions for Authors

Authors have well-addressed the raised issues. Now, it is suitable for publication.